


**The influence of sea- and land-breeze circulations on the diurnal variability of precipitation**
**over a tropical island**
Lei Zhu[1,2,3], Zhiyong Meng[1*], Fuqing Zhang[2,3], Paul M. Markowski[2]
*[1]Laboratory for Climate and Ocean-Atmosphere Studies, Department of Atmospheric and*
*Oceanic Sciences, School of Physics, Peking University, Beijing, China*
*[2]Department of Meteorology and Atmospheric Science, The Pennsylvania State University,*
*University Park, Pennsylvania*
*[3] Center for Advanced Data Assimilation and Predictability Techniques, The Pennsylvania State*
*University, University Park, Pennsylvania*

*Corresponding author address:* Dr. Zhiyong Meng, Laboratory for Climate and Ocean–
Atmosphere Studies, Department of Atmospheric and Oceanic Sciences, School of Physics, Peking
University, Beijing, China.

E-mail: zymeng@pku.edu.cn





**Abstract**

This study examines the diurnal variation of precipitation over Hainan Island in the South
China Sea using gauge observations from 1950 to 2010 and CMORPH satellite estimates from
2006 to 2015, as well as numerical simulations. Precipitation is most significant from April to
October, and exhibits a strong diurnal cycle resulting from land/sea breeze circulations. More than
60% of the total annual precipitation over the island is attributable to the diurnal cycle, with a
significant monthly variability as well. The CMORPH and gauge datasets agree well, except that
the CMORPH data underestimates precipitation and has a 1-h delay of peaks. The diurnal cycle of
the rainfall and the related land/sea breeze circulations during May and June were well captured
by convection-allowing numerical simulations with WRF, which were initiated from 10-year
average ERA-interim reanalysis, despite slightly overall overestimation and 1-h delay of the
rainfall peak. The diurnal precipitation is due to a diurnal cycle of moist convection, which initiates
around noontime owing to low-level convergence associated with the sea breeze circulation. The
precipitation intensifies rapidly thereafter and peaks in the afternoon with the collisions of sea
breeze fronts from different sides of the island. Cold pools of the convective storms contribute to
the inland propagation of the sea breeze. The precipitation dissipates quickly in the evening owing
to the cooling and stabilization of the lower troposphere and decrease of boundary-layer moisture.
Interestingly, the rather high island orography is not a dominant factor in the diurnal variation of
the precipitation over the island.





## 1. Introduction

On tropical islands, the diurnal precipitation cycle tends to be driven by the land-sea breeze
(LSB), as well as mountain-valley wind systems (Crosman and Horel 2010; Qian 2008; Mapes et
al. 2003).  Both rain gauge and satellite observations indicate that rainfall peaks in the late
afternoon over inland regions, and in the early morning or evening offshore (Yang and Slingo

2001).

The emergence of high temporal and spatial resolution satellite-estimated precipitation
observations, such as those provided by TRMM (Huffman et al. 2007) and CMORPH (Joyce et al.
2004), has greatly improved our understanding of tropical precipitation. Precipitation amounts are
much higher over tropical islands than their surrounding oceans (Qian 2008).  More than 34% of
the total precipitation in the tropics is attributable to precipitation over land (Ogrino et al. 2016).
Moreover, the precipitation is usually due to convection (Dai 2001), and tropical convection is
well known to have an important influence on the large-scale atmospheric circulation (Neale and
Slingo 2003; Sobel et al. 2011).
Many efforts have been made to understand the mechanisms behind the diurnal precipitation
cycle over tropical islands. Hassim et al. (2016) examined the diurnal cycle of rainfall over New
Guinea with a convection-allowing model.  They found that orography and the coastline along
with gravity waves were beneficial for the diurnal cycle. Other studies have found that
precipitation over tropical islands is strongly influenced by the size of the islands (Sobel et al. 2011;
Cronin et al. 2014).
The diurnal cycle of tropical rainfall is usually poorly captured by most global climate
models (GCMs), and even cloud-resolving models (CRMs), owing to model uncertainties in
depicting the physical mechanisms that underlie the diurnal precipitation cycle (Yang and Slingo





2001; Qian 2008; Nguyen et al. 2015; Hassim et al. 2016). Sometimes diurnal variability can only
be captured in some places or months where the signals are strong, while at other times, the diurnal
signals are captured, but with a large timing error of the maxima and minima. Studies show that
the LSB may have different contributions to the diurnal variabilities of precipitation at different
places (Yin et al. 2009; Jeong et al. 2011; Zhang et al. 2014; Zhang et al. 2016a, 2016b). Recent
studies (Bao et al. 2013; Chen et al. 2016, 2017) also have found that convectively driven cold
pools and latent cooling, as well as environmental wind and moisture, may play important roles in
the propagation and maintenance of diurnal rainfall in coastal regions. How cold pools and latent
cooling affect the diurnal cycle of rainfall and related LSB over a tropical island has not been
studied extensively.
This current work is aimed at examining the diurnal cycle of precipitation and the related
LSB over Hainan Island in the South China Sea. Hainan Island is a tropical island located off the
southern coast of China (Fig.1). It is one of the rainiest areas in China, and is influenced by a
variety of synoptic-scale and mesoscale weather systems, such as a monsoon, tropical cyclones,
LSB, and a mountain-plain solenoid (MPS). The island's topography features mountains in the
southwest, with peak altitudes of approximately 1000 m above the sea level (shaded in Fig. 1), and
plains in the northeast. Hainan Island is the largest of the so-called "Special Economic Zones" in
China. Tourism is an important part of Hainan's economy because of its beautiful beaches and
lush forests. Hainan Island is frequently referred to as "Chinese Hawaii." More than nine million
residents and tourists live on the island.
The characteristics of the precipitation and LSB over Hainan Island have been examined via
statistical methods based on either surface observation or modeling simulations (Tu et al. 1993;
Zhai et al. 1998; Zhang et al. 2014; Liang and Wang 2016). Based on nine station-based wind



observations, Zhang et al. (2014) found that LSB occurs rather frequently in summer and autumn,
though their findings are limited in using observations in only one month of one season. Most
recently, Liang and Wang (2016) examined the relationship between the sea breeze and
precipitation of Hainan Island using surface wind and precipitation observations along with the
global reanalysis over several years. They hypothesized that the seasonal precipitation is due to
the initiation of convection by the LSB, but they did not provide thorough investigation on how
the LSB circulations trigger and enhance the precipitation over Hainan Island.

The objective of this study is to investigate the diurnal precipitation variation over Hainan

Island and the detailed physical mechanisms related to LSB forcing and variability. The study
relies on rain gauge observations and satellite-derived precipitation estimates, as well as
convection-allowing numerical simulations. Section 2 describes the dataset and the methodology.
Section 3 documents the diurnal precipitation variation in each month, as well as the percentage
of the total precipitation that can be attributed to the diurnal cycle.  The relationship between the
precipitation and surface winds during the first rainy season [May and June, which is defined
relative to the second rainy season (July through September) of southern China when precipitation
is mainly caused by typhoons] also are analyzed. The model configuration and results of the
simulations are presented in Section 4. Conclusions are presented in Section 5.

## 2.  Observation dataset and methodology

Rainfall was analyzed using 19 rain gauges on Hainan Island (Fig. 1). The distribution of the

gauges is relatively homogeneous, and suitable for assessing the diurnal precipitation variation
over the entire island.  The dataset spans 60 years (1951–2010), though records exist for only a
subset of this period at some of the stations, owing to the fact that the stations were built at different





times.  Surface temperature and wind observations obtained at the same locations over a four-year
period (2007–2010) were used to investigate the land and sea breezes. The surface precipitation
observations were augmented by NOAA CMORPH analyses derived from low-Earth orbiting and
geostationary satellites (Joyce et al. 2004) as shown to be valuable in past studies of diurnal
precipitation over China (e.g., He and Zhang 2010; Bao et al. 2011; Sun and Zhang 2012; Bao et
al. 2013; Zhang et al. 2014). The spatial and temporal resolutions of the CMORPH analyses are
0.7277 degree by 0.7277 degree (approximately 8 km by 8 km) and 30 min, respectively.  Ten
years of CMORPH analyses (2006–2015) were used.

A series of convection-allowing numerical simulations were performed with the Weather

Research and Forecasting (WRF, Skamarock et al. 2008) model version 3.7.1 to investigate
dynamical features of the diurnal cycle of precipitation and its physical mechanisms, in particular
with regards to the LSB forcing. Initial and boundary conditions were provided by the European
Center for Medium-Range Weather forecast Interim (ERA-interim) reanalysis data (Dee et al. 2011)
of $0.75° \times 0.75°$ grid spacing. Only one domain was used with $225 \times 205$ grid points and a
horizontal grid spacing of 2 km. A total of 53 vertical levels were used with the model top at 10
hPa. The physical schemes that were used were the same as those in Chen et al. (2016), such as
the Yonsei University (YSU) boundary scheme (Hong et al. 2006) and the single-moment 5-class
microphysics scheme (Hong et al. 2004), except that no cumulus parameterization was used in this
study.

The initial conditions of all simulations were the average of ERA-interim reanalysis data at

0000 UTC in May and June of 2006–2015. The boundary conditions were obtained in the same
way as the initial conditions, cycled from 0000 to 0600, 1200, 1800 and 0000 UTC. A similar
methodology has been used to study diurnal cycle of precipitation in many different regions (Trier



et al. 2010; Sun and Zhang 2012; Bao and Zhang 2013; Chen et al. 2016, 2017). The biggest
advantage of this method is that it is able to capture the general characteristics of the diurnal cycle
of precipitation and the related dynamical processes instead of just focusing on a single case. All
simulations were integrated for one month.  The mean over the last 26 days was used for the
analyses in order to alleviate the spin-up issue and day-to-day variability.

**3. Observation analysis**
**3.1. Diurnal variation of precipitation and its seasonal-dependent features**

Diurnal variations of precipitation were examined at each single station in each month based

on the hourly gauge precipitation observation averaged over the period from 1951 to 2010. The
hourly precipitation evolution shows a significant seasonal cycle over the island. Most of the
precipitation falls from April to October and exhibits a distinct diurnal cycle during that period,
whereas less precipitation and lack of a strong diurnal cycle characterize the other months (Fig. 2).
The seasonal variability is related to the annual cycle of the East Asian Monsoon. April and
September are the two transitional periods of the low-level prevailing wind; the prevailing wind
strongly influences the transport of water vapor and precipitation.

The diurnal precipitation cycle has its maximum precipitation at 1500 local standard time

(LST, LST=UTC+8) in most months during the warm season, except at 1600 LST in April and
July. No second precipitation peak is observed, which is different from studies of other tropical
islands in which a second peak between midnight and early morning has been noted (Kishtawal
and Krishnamurti 2001; Wapler and Lane 2012; Chen et al. 2016). The second nocturnal peak was
found to be closely related to convection caused by the MPS that propagates offshore and coincides
with the land breeze during the night.



The diurnal precipitation cycle shows location-dependent features (Fig. 2). No distinct diurnal
variability of rainfall is observed at stations denoted by blue lines. These stations, denoted by blue
dots in Fig.1, are located along the southern coastline where there is no heavy and diurnal
precipitation. All the rest of the island stations share similar diurnal peak precipitation times with
the red-dot stations (in red lines) having the highest peak values from April to July, while in August
and September both the red-dot and black-dot stations (mostly inland) share the strongest peaks.
Even though the distribution of gauge-based precipitation stations is rather homogeneous, the
observations are still too sparse to analyze the rainfall pattern in detail over the island. For this
reason, satellite-derived precipitation CMORPH data also are used to examine the diurnal rainfall
variation for each of the 19 gauge stations. The hourly diurnal precipitation variation derived from
the CMORPH analyses agrees well with the rain gauge observations in each month (Fig. 3), though
the CMORPH amounts are slightly smaller. The time of peak precipitation in the warm season
(from May to August) is delayed by one hour in most months in the CMORPH analyses (maximum
at 1600 LST) relative to the peak in the gauge-based observations. These results indicate that the
CMORPH data are able to expose the diurnal precipitation cycle over this tropical island well in
comparison with the gauges, in particular for the warm-season months that are the focus of this
study.
The percentage of the diurnal precipitation (DP) in the total precipitation over the island in
each month was examined with the CMORPH data (Fig. 4). Similar to He and Zhang (2010) and
Bao et al. (2011), the diurnal precipitation percentage was defined as the mean rainfall rate at each
1-hour interval by DP $= \frac{\sum_{t=0}^{23} |(r_t - \bar{r})|}{r_d}$, in which, $r_t$ is the mean hourly precipitation at each hour $t$
(0–23), $\bar{r}$ is the mean hourly precipitation at all hours, and the $r_d$ is the daily mean precipitation.
The diurnal precipitation percentage represents a large percentage of the total precipitation over




the island in most months (Fig. 4). In particular, the total precipitation in May is almost entirely
attributable to the diurnal cycle (Fig. 4e). The diurnal contribution of the total precipitation exceeds
60% averaged for the whole year over the island, although the magnitude is smaller in September
and October. The diurnal precipitation percentage value exceeds 20% in August and September
(Figs. 4h and i). Moreover, the area exhibiting a large magnitude of diurnal precipitation roughly
coincides with the region also having the most accumulated precipitation. However, the diurnal
precipitation percentage is not quite related to the precipitation intensity. The precipitation is
extremely light in March and somewhat heavy in September. However, the diurnal precipitation
percentages are reversed (lesser percentages in September, higher percentages in March), which is
likely related to different physical processes of the precipitation in those months.

**3.2. The diurnal cycle of precipitation, land breezes, and sea breezes in May and June**

A more detailed analysis of the diurnal rainfall variation in May and June was carried out

because of the intense hourly mean rainfall and high DP percentage. In May, the prevailing warm
and wet southwest monsoon airflow transports abundant moisture from the ocean to Hainan Island.
A distinct diurnal cycle of precipitation, with a single peak between 1200 and 2000 LST, is evident
in both the gauge-based and CMORPH data (Fig. 5). The datasets agree well with each other at
each surface station, except that the CMORPH data exhibit larger peak values at the red and green
stations. Four gauge-based stations in blue have a much weaker daytime peak. These stations,
however, have an apparent nocturnal peak, whereas other stations do not exhibit a nocturnal peak.
The nocturnal precipitation is possibly attributable to the convergence between the low-level
prevailing wind and MPS circulations, which are to be examined with the numerical simulations





in section 4. The average over all stations (thick black line in Fig. 5) also exhibits an obvious
diurnal cycle.

The horizontal distribution of precipitation was analyzed using the CMORPH data (every 3 h

in Fig. 6) along with the perturbation surface wind at gauge stations, which was obtained by
subtracting the daily mean from the total wind to highlight the diurnal cycle. The precipitation
averaged over all times shows that the precipitation mainly appears in the northeast in the lee-side
of mountainous area (Fig. 6a). The gauge-based stations with significant diurnal cycle (in red dots)
are located over the heaviest rainfall region while the gauge-based stations with non-distinctive
diurnal feature (in blue dots) are located in the weakest precipitation area. Hourly variation of
precipitation shows that there is little precipitation over the island in the early to mid-morning
(0000 to 0900 LST), which is on average less than that over the surrounding ocean. At 0600 LST,
the perturbation surface wind at gauge stations has an offshore direction in coastal area, a signature
of nighttime land breeze (Fig. 6b). Three hours later at 0900 LST, the perturbation wind strengthens
and turns to the right of its previous direction, particularly along the coast (Fig. 6c). At 1200 LST,
the wind has changed to onshore direction as the beginning of sea breeze, along with the start of
weak inland precipitation where the sea breeze converges (Fig. 6d). In the next several hours (Figs.
6e–f), the rainfall intensifies rapidly, reaching to the peak at around 1700 LST. The heaviest
precipitation concentrates in the northeast island corresponding to strong convergence of sea
breeze (Fig. 6f). The precipitation dissipates rapidly thereafter and there is almost no precipitation
by 0300 LST (Figs. 6g–i). The perturbation wind also weakens quickly and turns to offshore along
the northern coast. The magnitude of the perturbation wind is close to zero over the island at 2100
LST (Fig. 6g). The land breeze intensifies slowly and nocturnal precipitation initiates along the
southeast coast of the island (Fig. 6h). The nocturnal precipitation intensifies to the peak and





expands to a larger area at 0300 LST, whereas the precipitation decreases to a minimum (near zero)
over the central island (Fig. 6i).

Although the analyses on the precipitation and surface wind observation can efficiently

reflect general features of the diurnal rainfall variation and the LSB, they cannot be used to
examine the detailed dynamics and thermodynamics processes of the diurnal precipitation cycle
and the related LSB over the tropical island. The three-dimensional structures of the LSB, as well
as the mechanism of how the LSB triggers and enhances the diurnal precipitation cannot be
resolved by the surface observation alone. These aspects were examined using a numerical model,
as discussed in the next session.

**4. Numerical simulations**

As described in the methodology section, all numerical simulations were initiated at 0000

UTC with the same diurnally cycled boundary conditions, both derived from a 10-year
climatological mean represented by the ERA-interim reanalysis for May and June during 2006–
2015. The initial conditions were modified for different purposes. Experiment REAL was initiated
with the unmodified initial conditions. Experiment NoTER is the same as REAL, except that the
orography over Hainan Island is removed in the initial conditions in order to isolate the influence
of the island's terrain.

**4.1. The simulated diurnal cycle and the influence of the orography**

The REAL simulation reproduces the diurnal cycle of precipitation and the associated LSB.

The diurnal variations of the 2-meter temperature, 2-meter temperature tendency, and precipitation
averaged over the last 26 days of the WRF simulations at all stations over the island (Fig. 7) show





generally good agreement with the observations except for slightly higher simulated 2-meter
temperature and greater simulated precipitation (cf. Figs. 7a and 7b). The overall process of the
diurnal variation over the island was well simulated, suggesting that the adopted numerical model
have ability to capture the radiative effect due to solar insolation well. The surface temperature
begins to increase at 0600 LST and peaks at 1300 LST, coincident with the increase of solar heating.
With the rainfall evaporation cooling rate becoming larger than the solar heating rate and/or the
radiative cooling later on, the temperature starts to decrease thereafter. After sunset, the
temperature drops continuously, reaching its minimum near 0600 LST.
The horizontal distribution of precipitation averaged in REAL (Fig. 8a) also has reasonably
good agreement with that of CMORPH at all hours (Fig. 6a), although the simulated precipitation
is slightly larger. The area of heavy precipitation at the center of the island is well captured by the
WRF simulation, although the magnitude is noticeably overestimated. The diurnal precipitation
cycle is also well revealed by the variation in the horizontal distribution of the simulated
precipitation although with a slightly larger magnitude and a 1-hour delay in peak time. The
evolution of the simulated surface perturbation wind (on the second lowest model level for
horizontal wind) is also consistent with the observation despite some discrepancy in magnitude
(Figs. 6 and 8), suggesting that the LSB is well captured as well.
The results of the NoTER simulation (with removal of island orography) are highly consistent
with those of REAL in terms of both the magnitude and timing of each variable averaged over the
whole analysis period and at all stations (Figs. 7b and c). Similar results are also found in the
horizontal distribution features (Figs. 8 and 9). Neither the pattern nor the magnitude is altered
meaningfully between the two simulations. These results suggest that the diurnal cycle
characteristics are not sensitive to the orography over Hainan Island, although many previous


studies demonstrated that the orography can play an important role in the precipitation over other
islands (Sobel et al. 2011; Hassim et al. 2016).
In order to simplify the influences of land category and coastline, experiment IDEAL was
further constructed with an idealized elliptical island to replace the real Hainan Island in the initial
condition. The idealized island has a similar size and orientation, and is located at the same place
as Hainan Island (Fig. 1), covered with uniform grassland while other areas of the model domain
are set as ocean. The diurnal variation of the 2-meter temperature (blue), 2-meter temperature
tendency (red) and hourly accumulated precipitation (green) in IDEAL (Fig. 7d) are nearly
identical to those in REAL (Fig. 7b) and the observation (Fig. 7a) except for their larger magnitudes
which could be related to the modified surface land category and the smoothed ellipsoidal
coastlines. The diurnal variations of the hourly accumulated precipitation and perturbation wind
on the second lowest model level for horizontal wind (Fig. 10) show that the timing of the LSB
transitions and the precipitation location are quite similar to those in REAL and the observation
with much smoother distribution in the horizontal perturbation wind and precipitation over the
island. The relationship between the diurnal variation of precipitation and LSB will be further
examined in details based on the results of IDEAL in the next section.

**4.2. Diurnal variation of precipitation and the related LSB in IDEAL**
The mean fields for averaged over all hourly model output times during the last 26 days of the
simulation depict a southerly low-level prevailing flow over the whole domain, which transports
warm moist air to the island from the South China Sea (Fig. 10a). Greater moisture appears in the
northern island over the heavy precipitation area under the influence of southwesterly low-to-mid-



level flow (850 hPa, Fig. 11a). Higher surface temperature appears over the southern side than that
in the northern side (Fig. 12a).
Based on the different phases of surface temperature and perturbation wind, we divided the
diurnal cycle process into four stages to elucidate the mechanisms in each stage. The four stages
are the establishment of a sea breeze (0600−1200 LST), peak sea breeze and peak precipitation
(1200−1800 LST), establishment of a land breeze (1800−0000 LST), and peak land breeze
(0000−0600 LST), respectively. More detailed analyses will be focused on the two middle stages.
These are the most complicated stages, but are also the most pertinent to the heavy diurnal
precipitation (and are therefore most interesting).

**a.  Stage 1. Establishment of a sea breeze (0600−1200 LST)**
This stage commences with the onset of surface heating following sunrise. Because ocean and
land have different heat capacities, the island is heated faster than the surrounding ocean. The
temperature gradient between the island and the surrounding ocean gradually reverses from
offshore to onshore, which results in the weakening and demise of land breeze, and the
establishment of a sea breeze over the island.
In the early morning hours when the sun is just about to rise, surface air temperatures over the
island attain their lowest readings, with air temperatures being a few degrees lower than over the
surrounding ocean (Fig. 12b). Owing to persistent solar heating, the surface air temperature over
most of the island exceeds that over the ocean by 0900 LST (Fig. 12c). Meanwhile, the surface air
temperature gradient is directed from offshore to onshore, although the land breeze still persists
over the island at this time.



The local rate of warming is inhomogeneous over the island. Surface temperatures in the
northeastern part of the island are considerably lower than that in other regions where the
temperatures surpass the surrounding ocean by 0900 LST. The slower warming in the northeastern
part of Hainan Island is likely due to the morning fog (Fig. 13b) that commonly forms overnight
within the area humidified by late-afternoon precipitation on the preceding day (Fig. 14b). The fog
attenuates solar radiation and subsequently slows the local warming.  The sea breeze begins to
develop along the southwestern coastline owing to the weakest land breeze and the highest
warming rate, while other areas of the island are still under the control of the land breeze with an
offshore temperature gradient (Fig. 12c). Two land-breeze circulations (LBCs) appear clearly in
the vertical direction below 3 km along the coast of the island at 0600 LST (Fig. 15b). The southern
LBC recedes quickly with the reversal of the temperature gradient at around 0900 LST, while the
other LBC remains distinct (Fig. 15c).
By 1200 LST, the temperature gradient reverses to the onshore direction, while the sea breeze
has fully established along the entire coastal line (Fig. 12d).  A sea-breeze front appears at the
leading edge of the sea breeze along the coastline, particularly along the northernmost coast where
the maximum near-surface temperature gradient lies (Fig. 12d). At the same time, copious water
vapor is transported inland from the ocean owing to the low-to-midlevel prevailing wind (Fig. 11d)
and upward motions (Fig. 14d). Clouds initially form along the sea-breeze front (Fig. 13c) and
subsequently produce rainfall (Fig. 10d).

**b. Stage 2. Peak sea breeze and peak precipitation (1200−1800 LST)**
Surface temperature is a maximum from 1200 to 1400 LST over most of the island, then
decreases rapidly thereafter owing primarily to the development of precipitation (which has its





diurnal maximum during this period) and associated evaporative cooling. The sea breeze also
reaches its peak intensity in the 1200–1400 LST time period.

At 1500 LST, surface temperature decreases over the rainfall area owing to evaporative

cooling, and slightly increases over other areas because of continuous solar heating (Fig. 12e).
There is significant enhancement in upward motions in the low to middle troposphere (Fig. 15e).
The sea breeze reaches its peak strength and greatest inland penetration (Fig. 12e). Two distinct
sea breeze circulations (SBCs) are clearly seen in the vertical cross section, with the stronger one
over the northern flank of the island (Fig. 15e). Moisture is transported to the northern part of the
island by the deep southwesterly prevailing wind throughout the lower troposphere (Fig.11e). At
the same time, enhanced vertical motions transport the low-level moisture to midlevels (Fig.14e).
These factors favor the development of deep convection over the northern flank of the island (Fig.
13e). As a result, precipitation increases significantly along the sea breeze front (Fig. 10e).

By 1800 LST, the strongest rainfall falls over the island (Fig. 10f) owing to strongest low-

level convergence and subsequent lifting of warm moist air (Fig. 14f). The sea breeze fronts move
further inland and collide with each other near the center of the island (Fig. 12f), with a deep layer
of moisture over the northern side of the island that fuels the strong precipitation (Figs. 11f and
14f). Cold pools form due to the evaporative cooling of the precipitation, contributing to the
formation and organization of new convection, which further adds to the precipitation. The
precipitation pattern (Fig. 10f) exhibits a horseshoe shape aligned with the prevailing wind
direction, which is similar to the result of the urban heat island study by Han and Baik (2008).

**c. Stage 3. Establishment of a land breeze (1800–0000 LST)**



During this period, the convection quickly dissipates and the sea breeze is replaced by a land
breeze (Figs. 10g and h) after sunset. The surface temperature decreases continuously throughout
this stage over the island. The rate of temperature decrease is fastest in the first several hours (Fig.
7d) due to the sudden loss of solar heating. The horizontal temperature gradient begins to reverse,
which eventually leads to the establishment of the land breeze (Figs. 12g and h). By 2100 LST,
approximately two hours after sunset, temperature over the island is decreasing rapidly both at the
surface (Fig. 12g) and throughout the boundary layer (Fig. 15g). Meanwhile, subsidence becomes
dominant over the island (Fig. 15g). The subsidence dries the lower levels and rainfall has ceased
over the whole island (Fig. 14g).
By 0000 LST, with the continuous decreasing of temperature and amplifying of the offshore
temperature gradient, the land breeze circulations are well established in particular across the shore
of the northern island (Fig. 15h). Further drying is seen in mid-to-low levels (Figs. 11h and 14h).
Clouds vanish quickly and precipitation dissipates almost completely by this time (Fig. 13h).

**d. Stage 4. Peak land breeze (0000–0600 LST)**
The land breeze reaches its maximum intensity during this period. Nighttime radiative cooling
results in the minimum temperature being attained at approximately 0500 LST. From 0000 to 0300
LST, the land breeze intensifies rapidly along the northwest coast, becoming nearly perpendicular
to the coastline and parallel with the low-level prevailing wind as the surface temperature over
land decreases (Fig. 12i). Two LBCs are evident in the vertical cross section (Fig. 15i). Subsidence
extends over the entire island (Fig. 14i). The peak land breeze is established at 0600 LST  (Fig.
12b). The strong subsidence also leads to further midlevel drying (Fig. 11b). Near the surface, the



cooling is associated with an increase in the relative humidity, which leads to the formation of low
clouds and fog (Figs. 13b and 14b).

### 4.3. The impacts of latent heating/cooling on the LSB and related rainfall

In order to examine the impact of latent heating/cooling on the LSB and related rainfall, a
"FakeDry" simulation was performed similar to the IDEAL experiment, except for turning off
latent heating and cooling in the model. Surface temperature in the FakeDry experiment agrees
well with the IDEAL over the island (cf. Figs. 12 and 16), which indicates that the solar heating
rather than the latent heating/cooling is primarily responsible for the temperature variability.
Although the precipitation is decreased significantly (cf. Figs. 10 and 17), light rainfall still occurs
in the late afternoon in conjunction with the sea breeze front, but with an approximate 3-h delay
in convection initiation. The precipitation attains its maximum at 1800 LST, which along with the
peak sea breeze, also lags that in the IDEAL experiment by approximately three hours (cf. Figs.
10e and 17f).
The impact of cold pool and latent cooling on the sea breeze and rainfall was further examined
using the Hovmöller diagrams of zonal wind perturbation on the second lowest model level for
horizontal wind and hourly precipitation along the red line in Fig. 1 for both experiments IDEAL
and FakeDry (Fig. 18). A weaker sea breeze is observed in the FakeDry experiment than in the
IDEAL experiment. The propagation of the LSB is much slower and the inland propagation
distance is much shorter than that in the IDEAL experiment, which suggests that the cold pool can
accelerate the propagation and intensification of the sea breeze over the tropical island. Moreover,
given precipitation varies precisely with the convergence and divergence of horizontal winds due



to LSB in both simulations, it is evident that the LSB is the primarily forcing for the diurnal
precipitation variability over the island.
The LSB circulations in the FakeDry experiment are similar to those in the IDEAL experiment,
except that they are confined to lower levels owing to weaker vertical motion (Fig. 19). The latent
heating can strengthen vertical motions and extend the LSB circulations to higher altitudes. The
latent heating feedback can also lead to stronger and earlier convection initiation and precipitation
along the sea-breeze fronts. In turn, the cold pool further promotes the inland penetration of the
sea-breeze front and enhances the precipitation (cf. Figs. 18a and 18b).

**405 5. Summary**

This study explored the diurnal precipitation variation and its relationship with the land/sea
breeze circulations on Hainan Island, a tropical island located off the southern coast of China,
based on gauge observation and satellite-estimated precipitation, as well as convection-allowing
numerical simulations. The diurnal cycle of precipitation in each month over the island was
analyzed with 19 gauge observations during 1951–2010. Most precipitation fell during the warm
season (from April to October) and exhibited a significant diurnal cycle, whereas much lesser
precipitation fell in other months. Precipitation is a maximum between 1500–1700 LST in the
warm season at almost all stations except for four stations along the southern coastline of the island.
The satellite-derived CMORPH precipitation estimates from 2006–2015 were further used to
validate the diurnal precipitation variation. The CMORPH data agrees well with gauge
observations except for a smaller magnitude of precipitation and a 1-hour delay in the timing of
the daily precipitation maximum during the warm season. The CMORPH data analyses show that
about 60% of the total annual precipitation over the island is attributable to diurnal variations, with



the largest proportion in May and the smallest proportion in September and October. For May and
June, precipitation begins around local noon time, intensifying quickly thereafter, and reaching a
peak at ~1500 LST based on station observations. This diurnal rainfall cycle is, for the most part,
consistent with the diurnally varying low-level wind convergence and divergence.

A series of numerical simulations using a convection-allowing configuration of the WRF

model (2-km horizontal grid spacing) were conducted to understand the underlying mechanisms
of the diurnal precipitation variations. The initial and cyclic boundary conditions were generated
using a 10-year (2006–2015) average of ERA-interim data for May and June. It is found that the
orography of Hainan Island may be of only secondary influence on the diurnal precipitation cycle,
which is different from past studies on other hilly islands. Similar diurnal cycles of precipitation
and related land/sea breeze circulations were simulated between simulations with and without
orography over the island.  Even with an idealized island, which is an elliptical flat island located
at the same place with similar area and orientation, but only grassland land cover, the diurnal cycle
characteristics can still be fairly well captured.

The simulated diurnal cycle of precipitation and related land/sea breeze circulations based on

the idealized flat-island simulation were divided into four stages in terms of the evolutions of
temperature, winds and precipitation. Stage 1 is from 0600 to 1200 LST, during which time the
land breeze is replaced by a sea breeze as solar heating warms the interior of the island. Abundant
moisture is transported to the low to middle troposphere over the island, resulting in convection
initiation and precipitation along the sea-breeze front. Stage 2 is from 1200 to 1800 LST, during
which time sea breeze attains its peak intensity and precipitation is a maximum. The sea breezes
from opposite sides of the island eventually penetrate all the way to the island's center and collide,
which results in the maximum precipitation being located there. Stage 3 is from 1800 to 0000 LST,



during which time a land breeze is established as a result of cooling over the island. The cooling
is due primarily to the sudden loss of solar heating. Subsidence from the land breeze prevents
further precipitation by early evening. The last stage covers the peak land breeze, which is
observed near sunrise.
The FakeDry experiment shows that the latent cooling and cold pool have a small impact on
the land/sea breeze circulations but can apparently enhance precipitation. Strong convection can
enhance the sea breeze, and the augmentation of the sea breeze by the evaporatively driven cold
pool helps to accelerate the inland propagation of the sea breeze.
Finally, it is worth mentioning that the 1-hour delay in the timing of the maximum
precipitation in the simulation is probably caused by the 2-km horizontal resolution, which may
not be high enough to resolve explicit underlying physical process. It is likely for the same reason
that the weak nocturnal precipitation is not captured by the simulations. Much higher horizontal
and vertical resolution might be needed in the future work to resolve more detailed processes
related to the diurnal rainfall cycles.

**Acknowledgments:** Lei Zhu is supported by the Natural Science Foundation of China Grant
41461164006, and the Chinese Scholarship Council (CSC). Zhiyong Meng is supported by the
Natural Science Foundation of China Grants 41461164006, 41425018 and 41375048. Fuqing
Zhang is supported by the Office of Naval Research Grant N000140910526 and the National
Science Foundation Grant AGS-1305798. Paul Markowski is supported by National Science
Foundation grant AGS-1536460 and National Oceanic and Atmospheric Administration awards
NA15NWS4680012 and NA14NWS4680015. The simulations were performed on the Stampede
supercomputer of the Texas Advanced Computing Center (TACC). All data are freely available



from sources indicated in the text or from the corresponding author upon request (Email:
zymeng@pku.edu.cn).



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





**Figure Captions**

FIG. 1. Configuration of model domain, gauge-based station points (color dots correspond to the
time series shown in Fig. 2) over Hainan Island and the terrain height (shading, m). The red ellipse
is the idealized representation of the island (used for the idealized simulations), and the red vertical
line indicates the location of the vertical cross-sections shown in Figs. 14–16.

FIG. 2. Average rainfall accumulations by hour, each month of the year, obtained from the rain
gauge network. The color is consistent with the color dots over the island in Fig. 1. LST means the
Local Standard Time.

FIG. 3. Average station rainfall accumulations obtained from gauges (blue) and CMORPH (red)
in each month.

FIG. 4. Fraction of the total precipitation that can be attributed to the diurnal cycle, by month
(shading), along with average hourly precipitation accumulations (black contours every 0.05 mm,
starting at 0.25 mm).

FIG. 5. Average rainfall accumulations by hour in May and June (a) from rain gauges and (b)
derived from CMORPH.

FIG. 6. Ten-year average, hourly rainfall accumulations at 3-h intervals for May and June derived
from CMORPH (shading) except used 1700 LST as it is the strongest rainfall time in CMORPH
observation. Three-year average wind velocity (vectors) is also shown. Rain gauge locations are
indicated in (a).

FIG. 7. The average of 2-meter temperature (T2_avg), 2-meter temperature tendency
(T2_tendency, temperature difference between two neighboring hours), and hourly rainfall
accumulation over the island based on (a) gauge observations, (b) simulation REAL, (c) simulation
NoTER, (d) and simulation IDEAL. Horizontal colored lines indicate means over all hours.

FIG. 8. Hourly precipitation accumulation (shading) and average perturbation wind (vectors) on
the second lowest model level for horizontal wind in simulation REAL every 3 h. The averages
over all hours are shown in (a).

FIG. 9. As in FIG. 8, but for simulation NoTER.

FIG. 10. As in FIG. 8, but for simulation IDEAL.

FIG. 11. Water vapor mixing ratio (shading) and horizontal wind (vectors) at 850 hPa, and hourly
precipitation accumulations > 0.1 mm (thick purple contours), (b–i) every 3 h and (a) averaged
over all times.

FIG. 12. (a) 2-meter mean temperature (shading) and horizontal wind (vectors) on the second
lowest model level for horizontal wind; (b–i) 2-meter mean temperature perturbation (shading)



and mean perturbation horizontal wind (vectors) on the second lowest model level every 3 h. The
right color bar is used for (a).
FIG. 13. Cloud water mixing ratio (red shading), 2-meter temperature (grey shaded), perturbation
horizontal wind on the second lowest model level for horizontal wind (yellow vectors), and hourly
precipitation accumulation (green contour lines) every 3 h.
FIG. 14. Vertical cross-sections of water vapor mixing ratio (shading), perturbation wind (vectors;
the scale of the vertical component is increased by a factor of 5), and temperature (contours) in the
south-to-north direction (see red line in Fig. 1) averaged over all hours (a) and at 3-h intervals (b–
i). The triangles in each panel indicate the edges of the island.
FIG. 15. Vertical cross-sections of perturbation temperature (shading), perturbation wind (vectors;
the scale of the vertical component is increased by a factor of 5), and temperature (contours) in the
south-to-north direction (see red line in Fig. 1) averaged over all hours (a) and at 3-h intervals (b–
i). The triangles in each panel indicate the edges of the island.
Fig. 16. As in Fig. 12, but for simulation Fakedry.
Fig. 17. As in Fig. 8, but for simulation Fakedry.
Fig. 18. Hovmoller diagrams of perturbation meridional wind component on the second lowest
model level for horizontal wind (shading) in the (a) IDEAL and (b) Fakedry simulations,
respectively. Precipitation exceeding 0.1 mm is enclosed by the heavy purple contours. The two
vertical dash lines indicate the edges of the island.
Fig. 19. As in Fig. 15, but for simulation Fakedry.






**Figures**

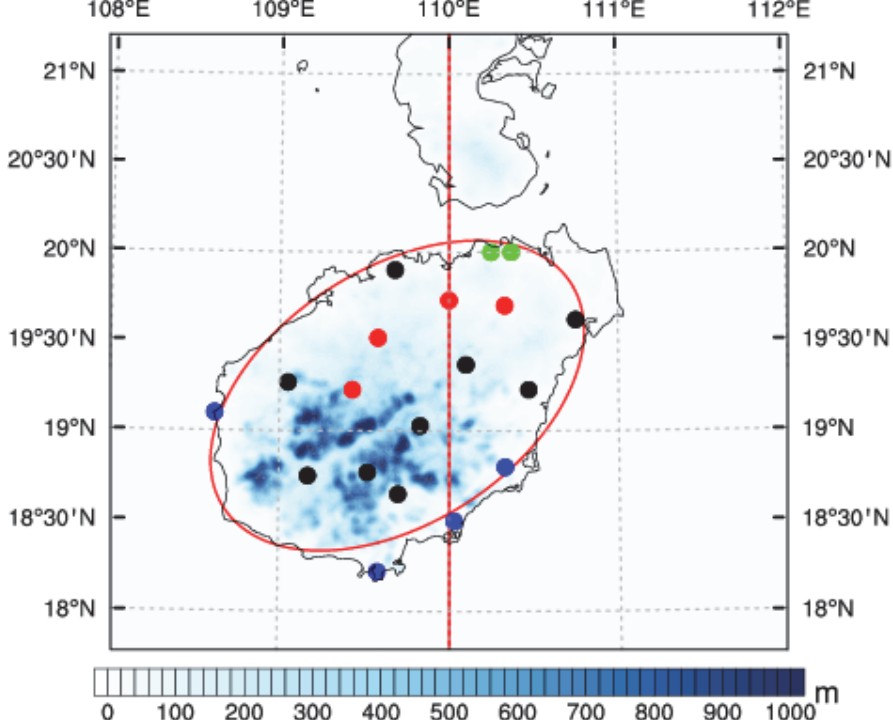

FIG. 1. Configuration of model domain, gauge-based station points (color dots correspond to the
time series shown in Fig. 2) over Hainan Island and the terrain height (shading, m). The red ellipse
is the idealized representation of the island (used for the idealized simulations), and the red vertical
line indicates the location of the vertical cross-sections shown in Figs. 14–16.





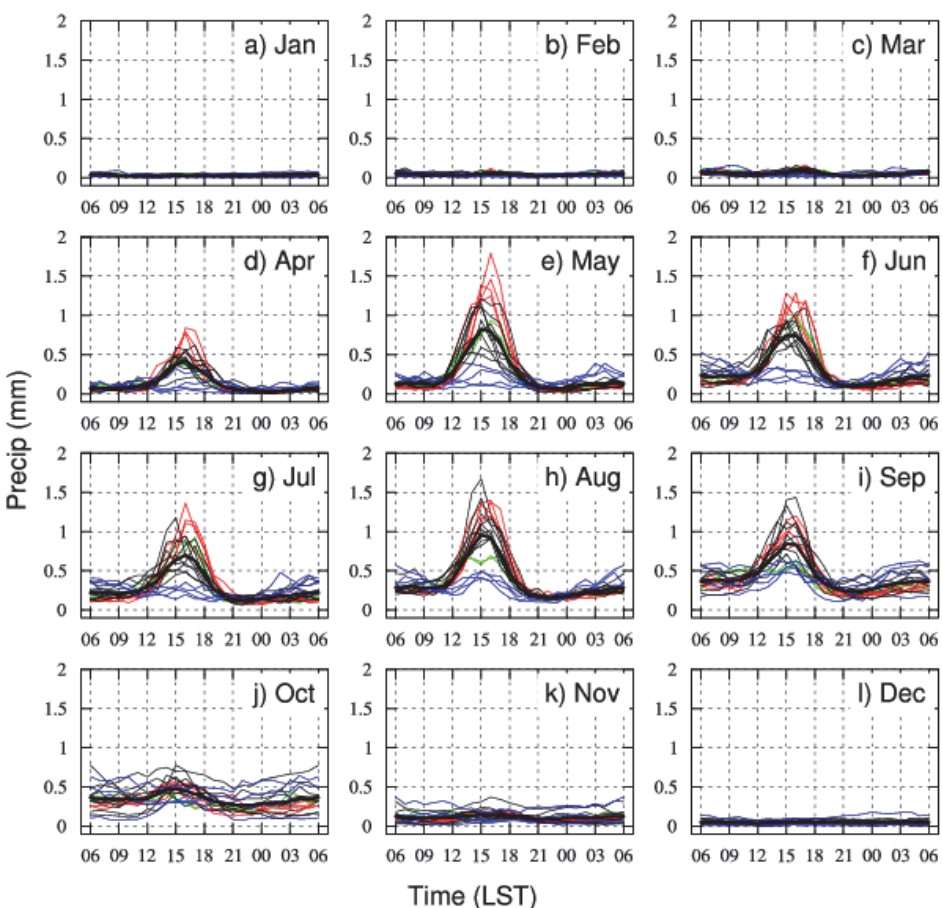

FIG. 2. Average rainfall accumulations by hour, each month of the year, obtained from the rain
gauge network. The color is consistent with the color dots over the island in Fig. 1. LST means the
Local Standard Time.





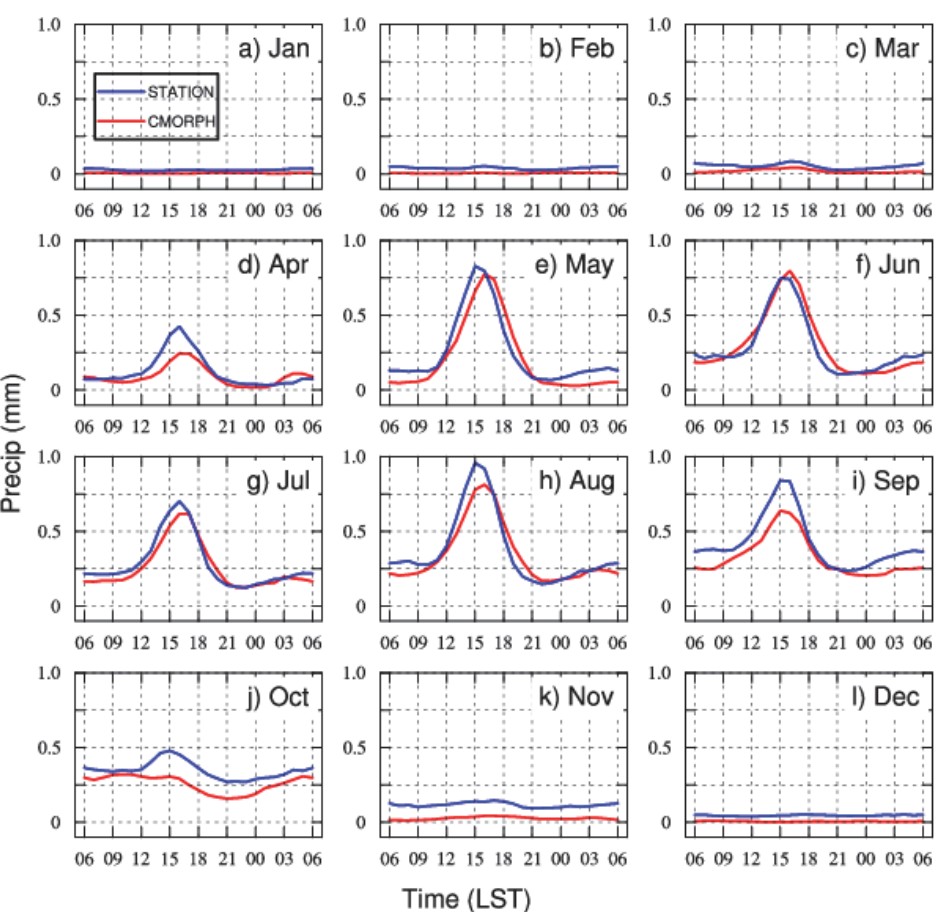

FIG. 3. Average station rainfall accumulations obtained from gauges (blue) and CMORPH (red) in each month.






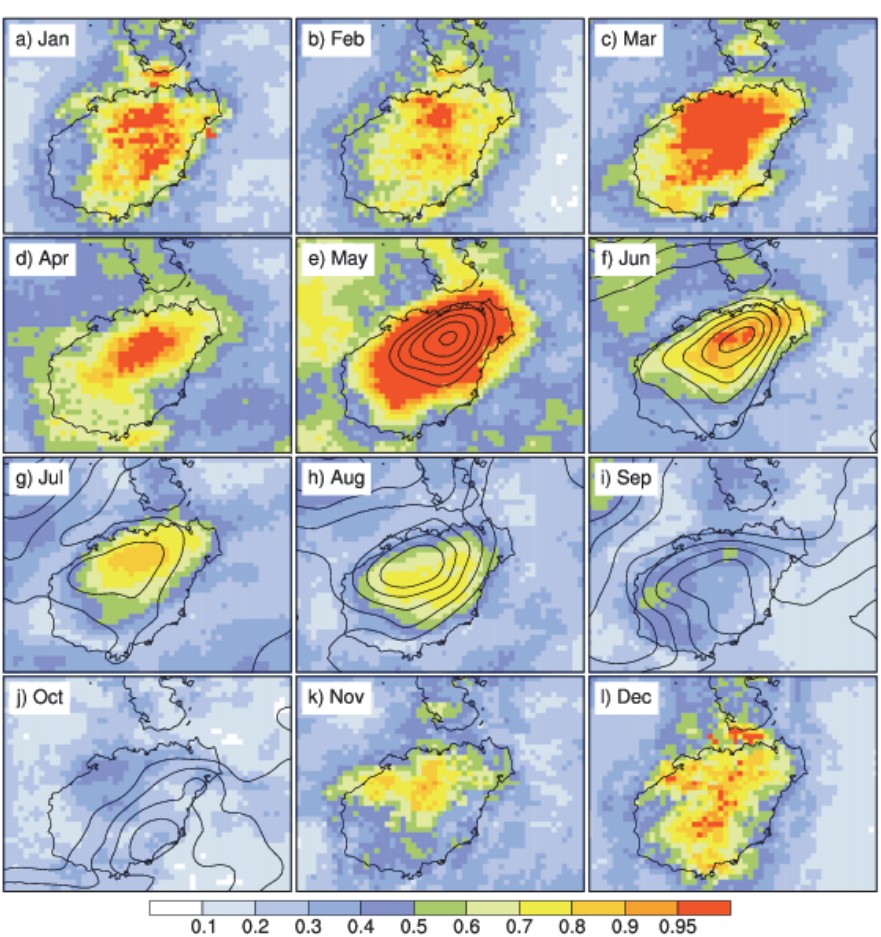

FIG. 4. Fraction of the total precipitation that can be attributed to the diurnal cycle, by month
(shading), along with average hourly precipitation accumulations (black contours every 0.05 mm,
starting at 0.25 mm).




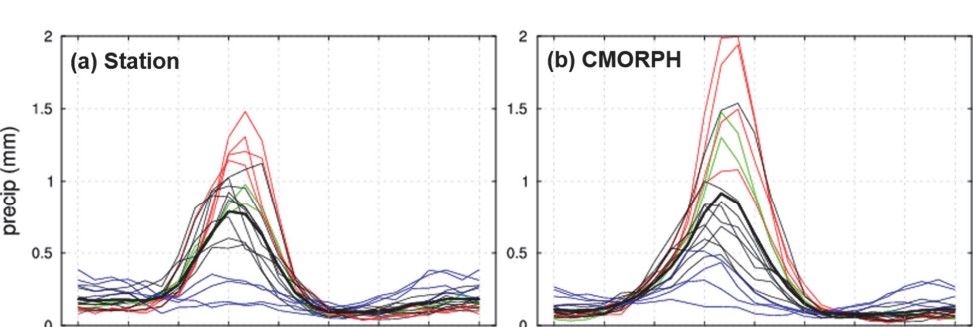

FIG. 5. Average rainfall accumulations by hour in May and June (a) from rain gauges and (b)
derived from CMORPH.





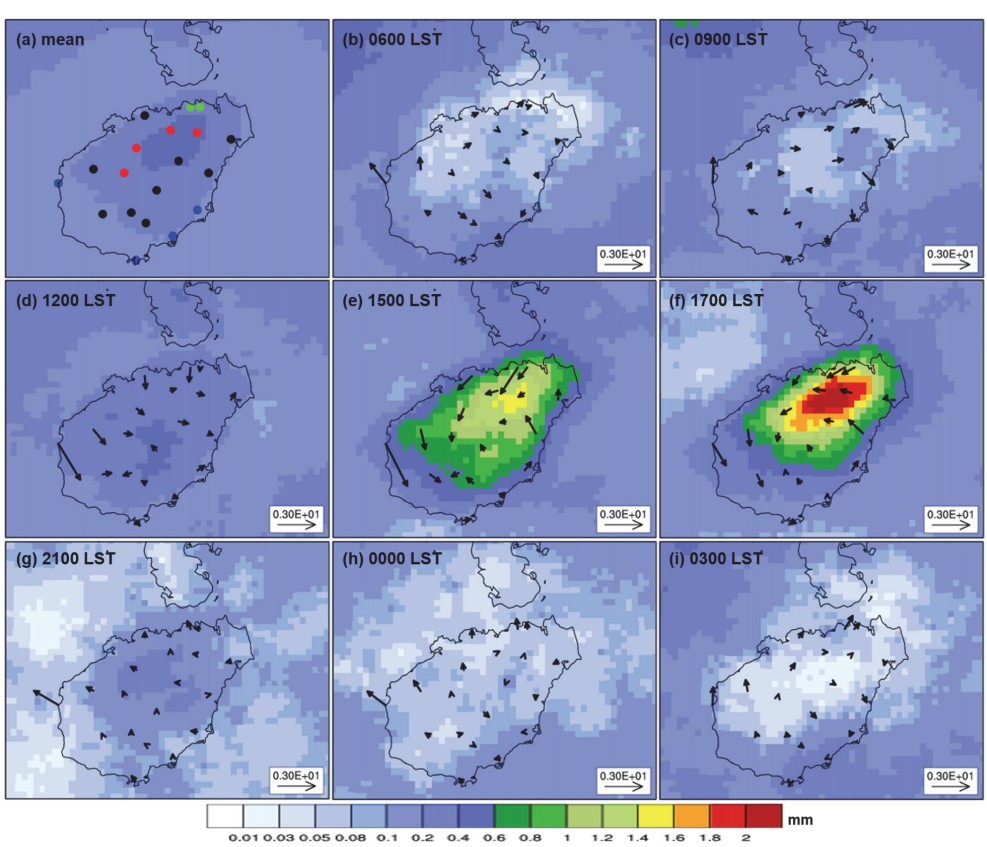

FIG. 6. Ten-year average, hourly rainfall accumulations at 3-h intervals for May and June derived
from CMORPH (shading) except used 1700 LST as it is the strongest rainfall time in CMORPH
observation. Three-year average wind velocity (vectors) is also shown. Rain gauge locations are
indicated in (a).






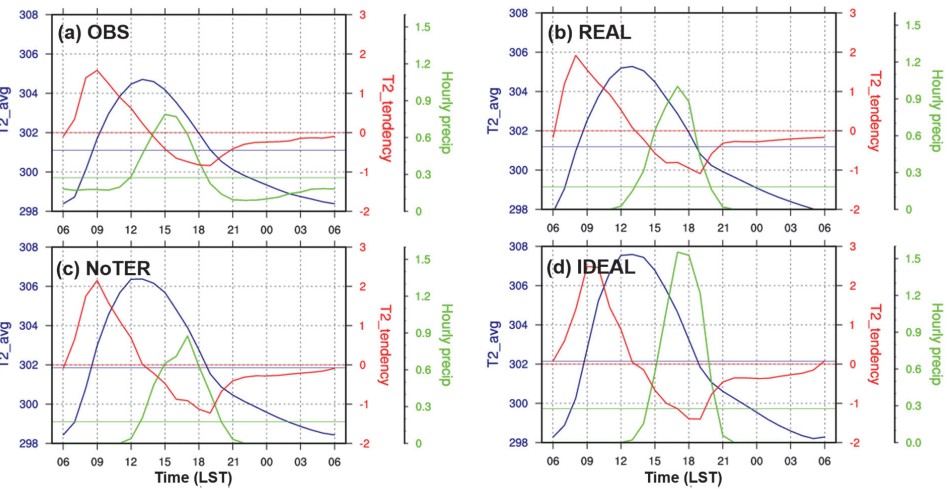

FIG. 7. The average of 2-meter temperature (T2_avg), 2-meter temperature tendency
(T2_tendency, temperature difference between two neighboring hours), and hourly rainfall
accumulation over the island based on (a) gauge observations, (b) simulation REAL, (c) simulation
NoTER, (d) and simulation IDEAL. Horizontal colored lines indicate means over all hours.






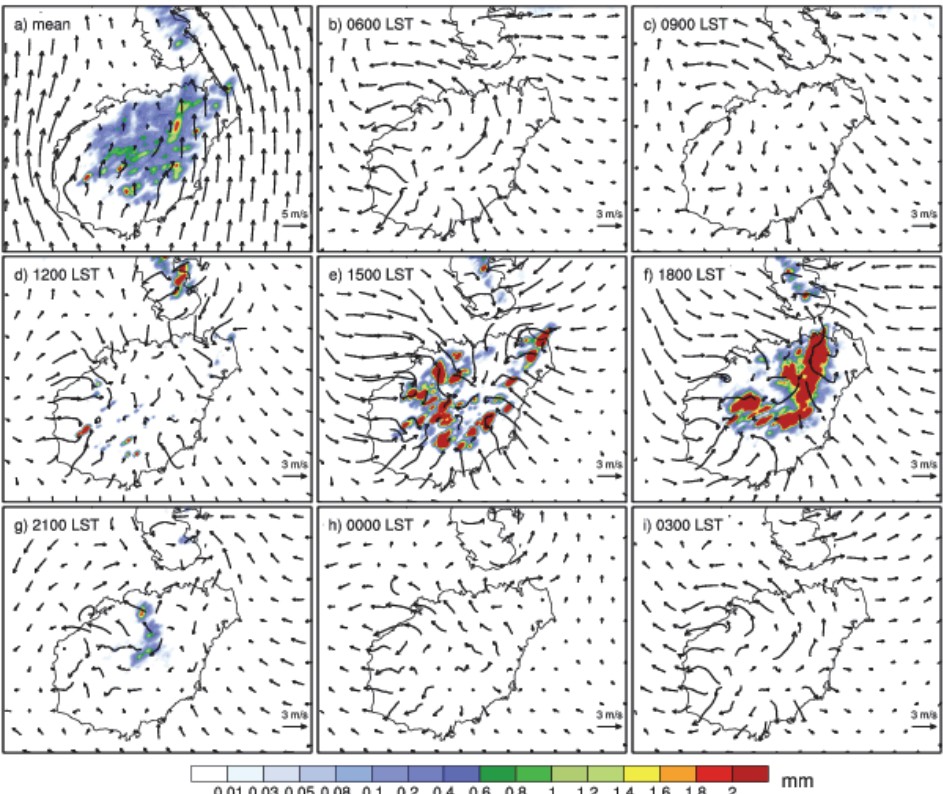


FIG. 8. Hourly precipitation accumulation (shading) and average perturbation wind (vectors) on
the second lowest model level for horizontal wind in simulation REAL every 3 h. The averages
over all hours are shown in (a).






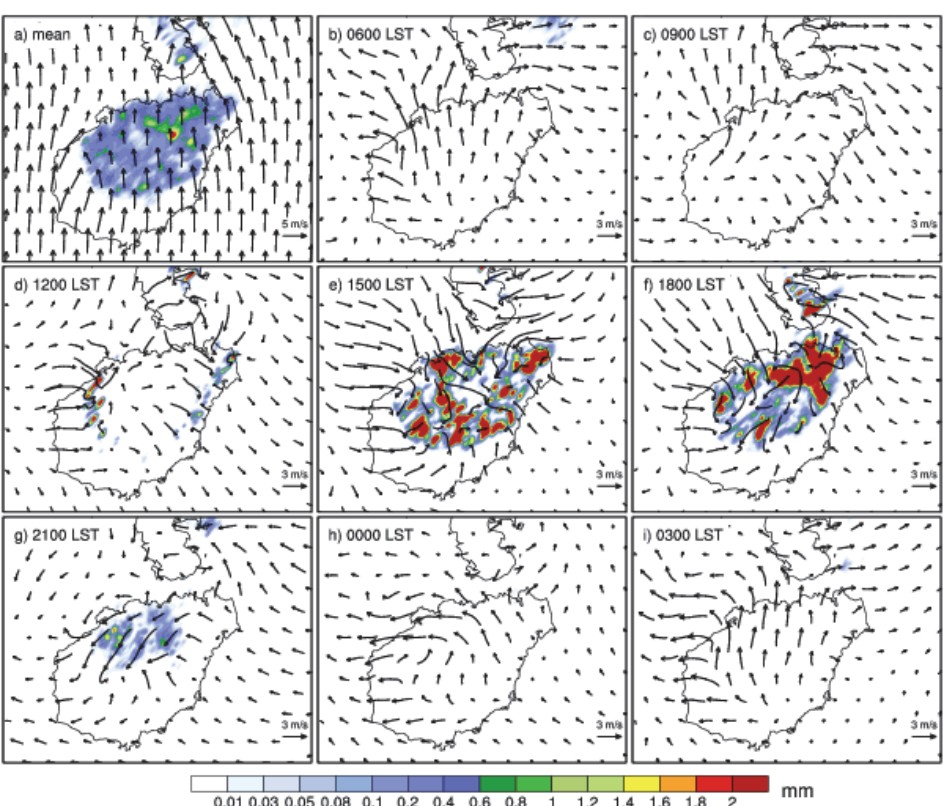

FIG. 9. As in FIG. 8, but for simulation NoTER.




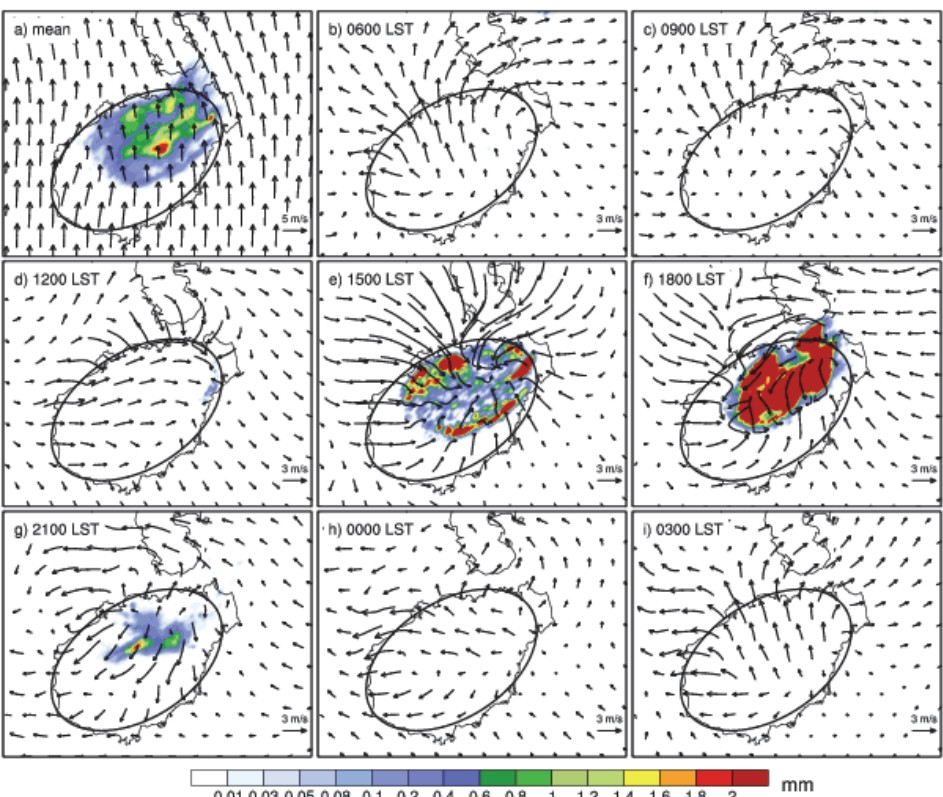

FIG. 10. As in FIG. 8, but for simulation IDEAL.




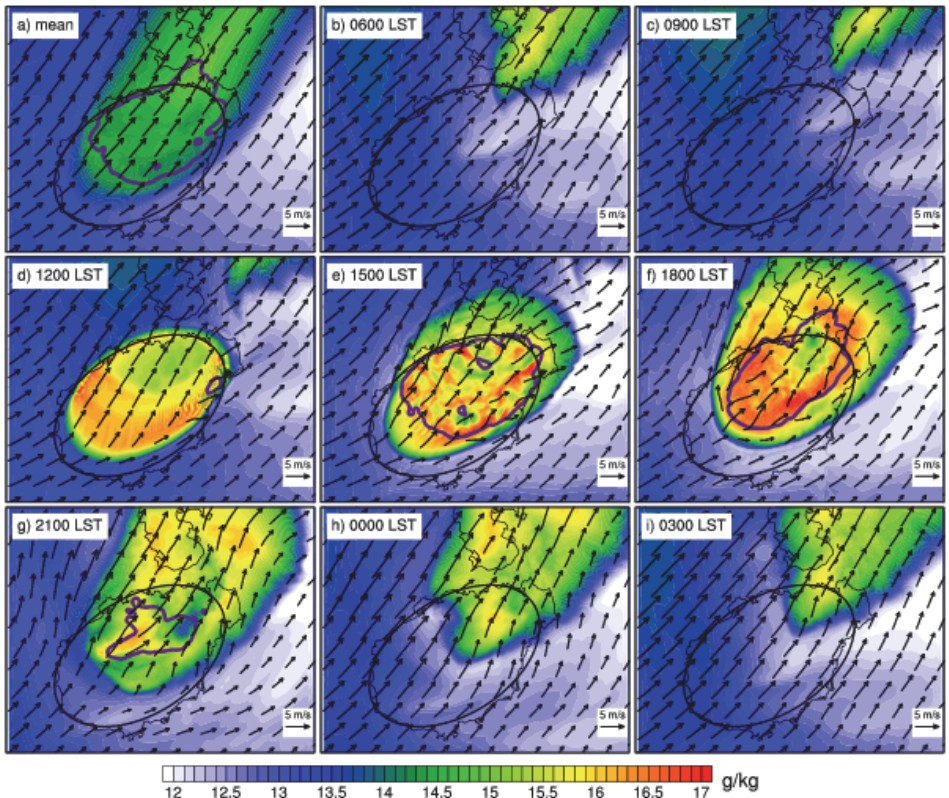

FIG. 11. Water vapor mixing ratio (shading) and horizontal wind (vectors) at 850 hPa, and hourly
precipitation accumulations > 0.1 mm (thick purple contours), (b–i) every 3 h and (a) averaged
over all times.




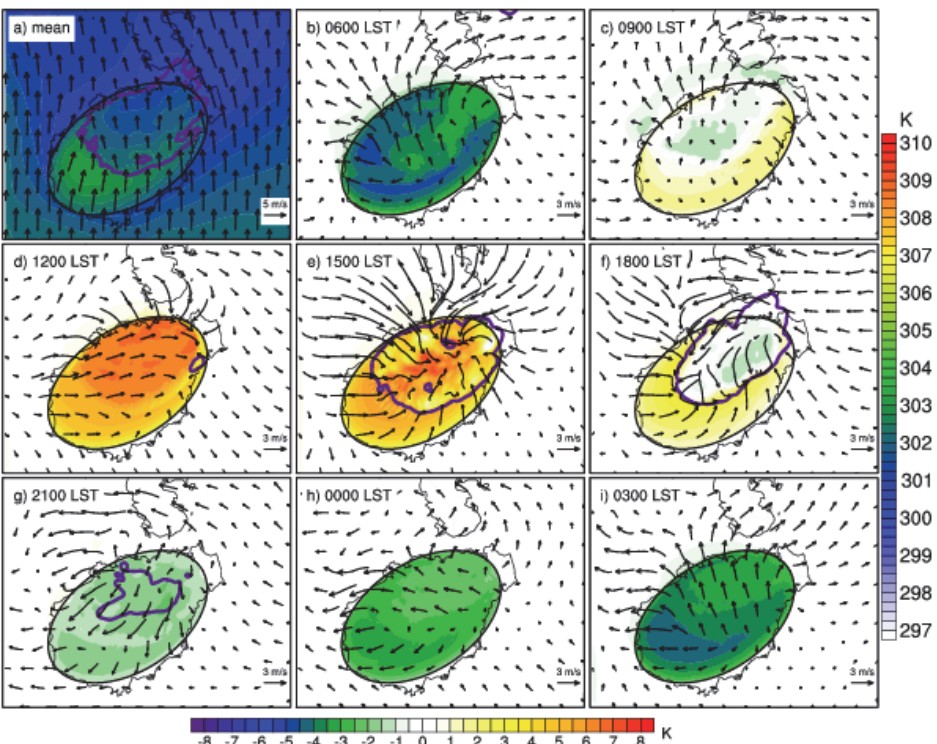

FIG. 12. (a) 2-meter mean temperature (shading) and horizontal wind (vectors) on the second
lowest model level for horizontal wind; (b–i) 2-meter mean temperature perturbation (shading)
and mean perturbation horizontal wind (vectors) on the second lowest model level every 3 h. The
right color bar is used for (a).



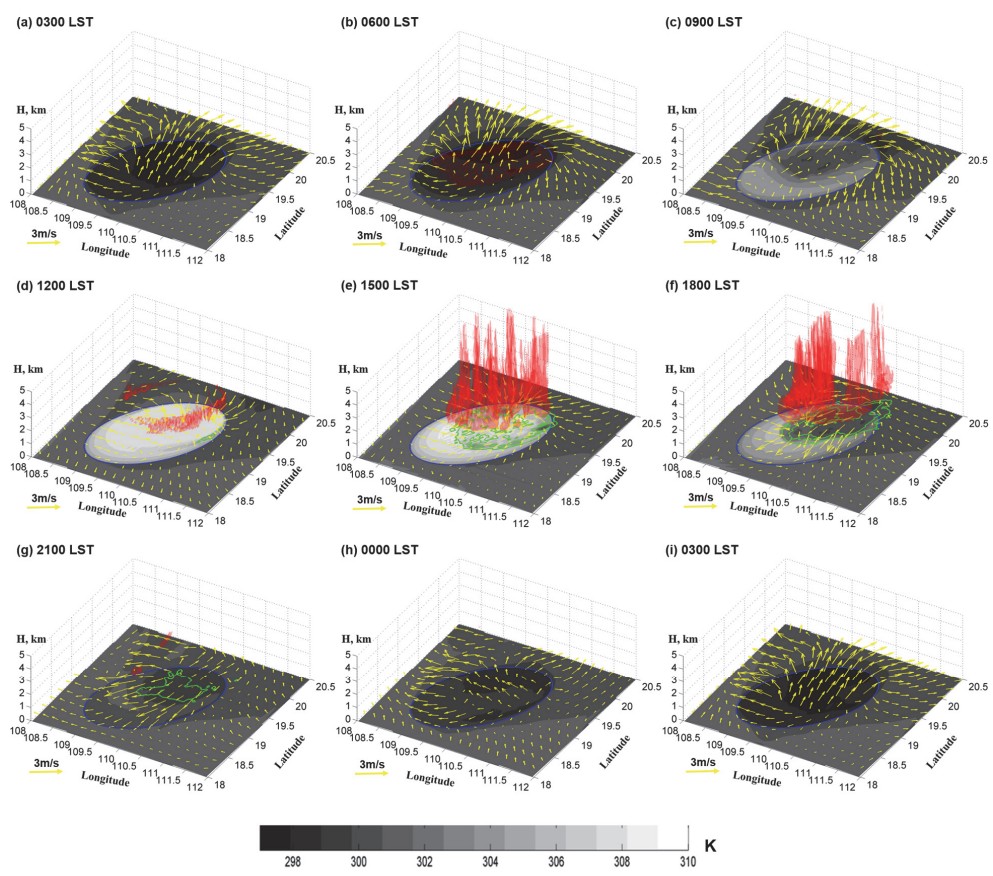

FIG. 13. Cloud water mixing ratio (red shading), 2-meter temperature (grey shaded), perturbation
horizontal wind on the second lowest model level for horizontal wind (yellow vectors), and hourly
precipitation accumulation (green contour lines) every 3 h.





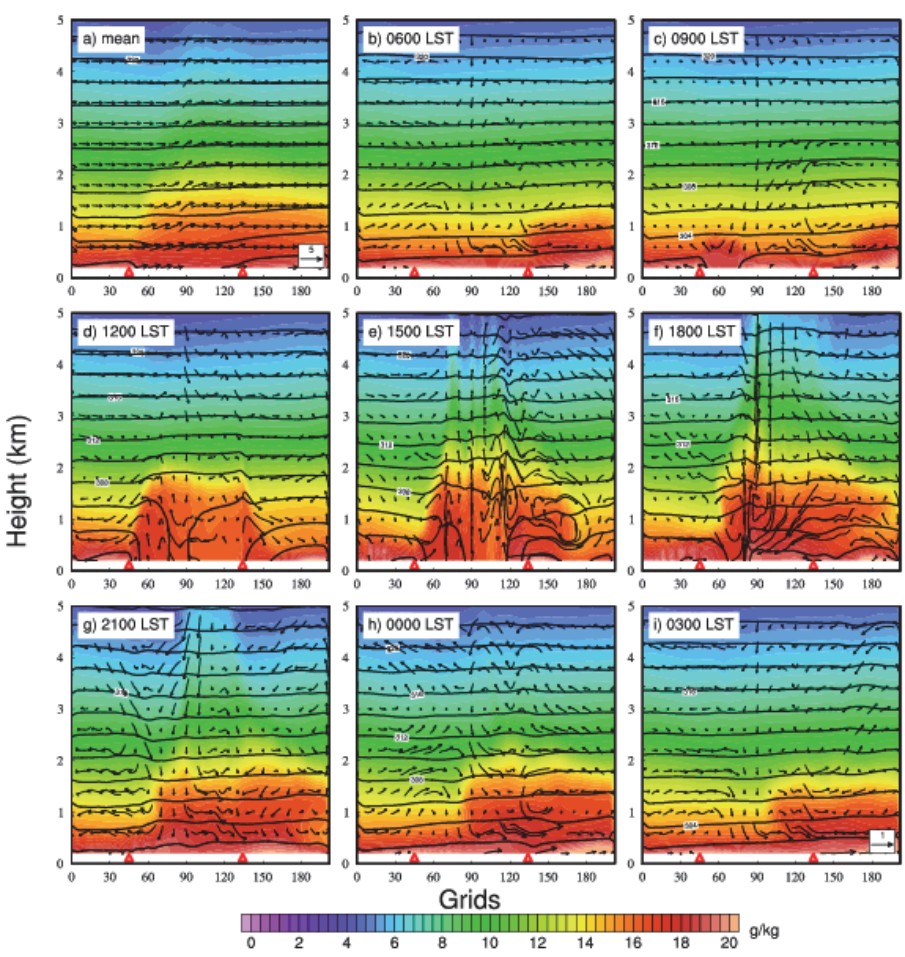

FIG. 14. Vertical cross-sections of water vapor mixing ratio (shading), perturbation wind (vectors;
the scale of the vertical component is increased by a factor of 5), and temperature (contours) in the
south-to-north direction (see red line in Fig. 1) averaged over all hours (a) and at 3-h intervals (b–
i). The triangles in each panel indicate the edges of the island.





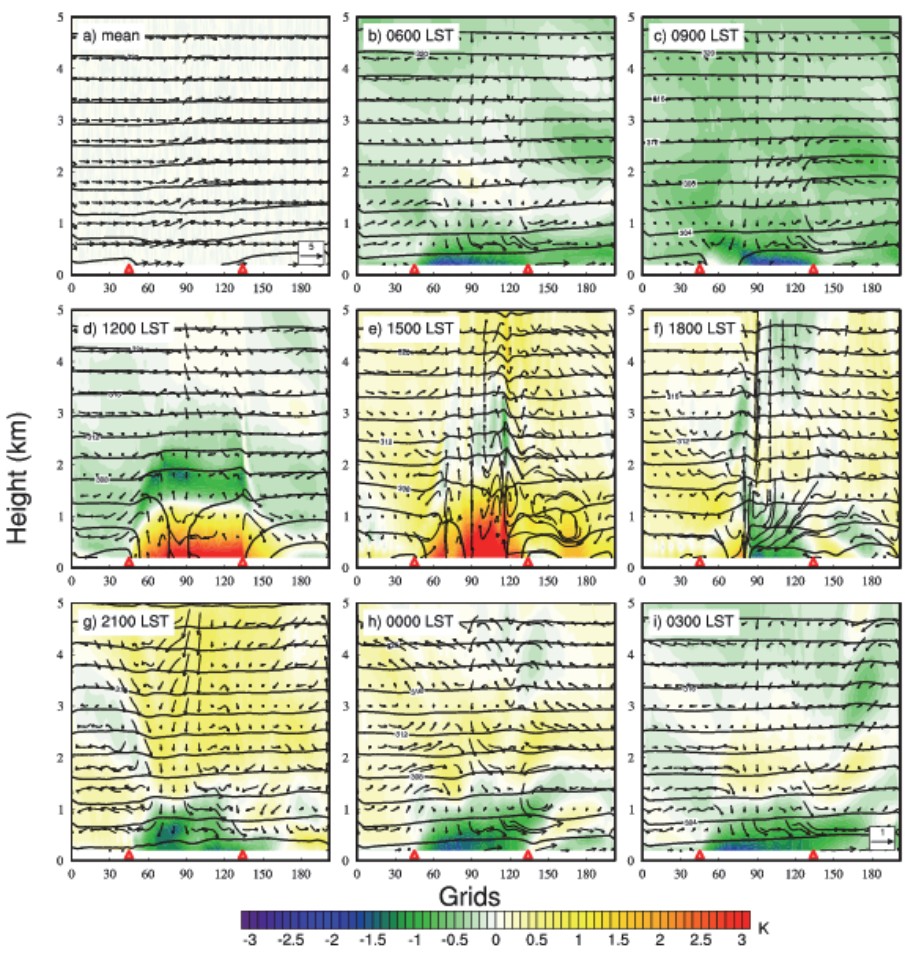

FIG. 15. Vertical cross-sections of perturbation temperature (shading), perturbation wind (vectors;
the scale of the vertical component is increased by a factor of 5), and temperature (contours) in the
south-to-north direction (see red line in Fig. 1) averaged over all hours (a) and at 3-h intervals (b–
i). The triangles in each panel indicate the edges of the island.




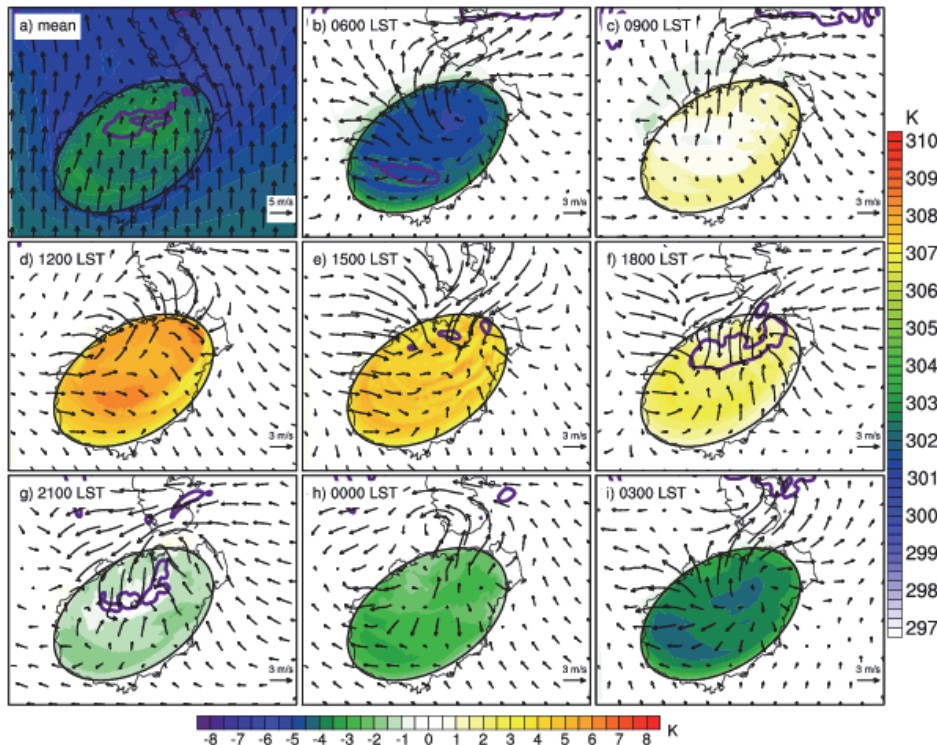

Fig. 16. As in Fig. 12, but for simulation Fakedry.





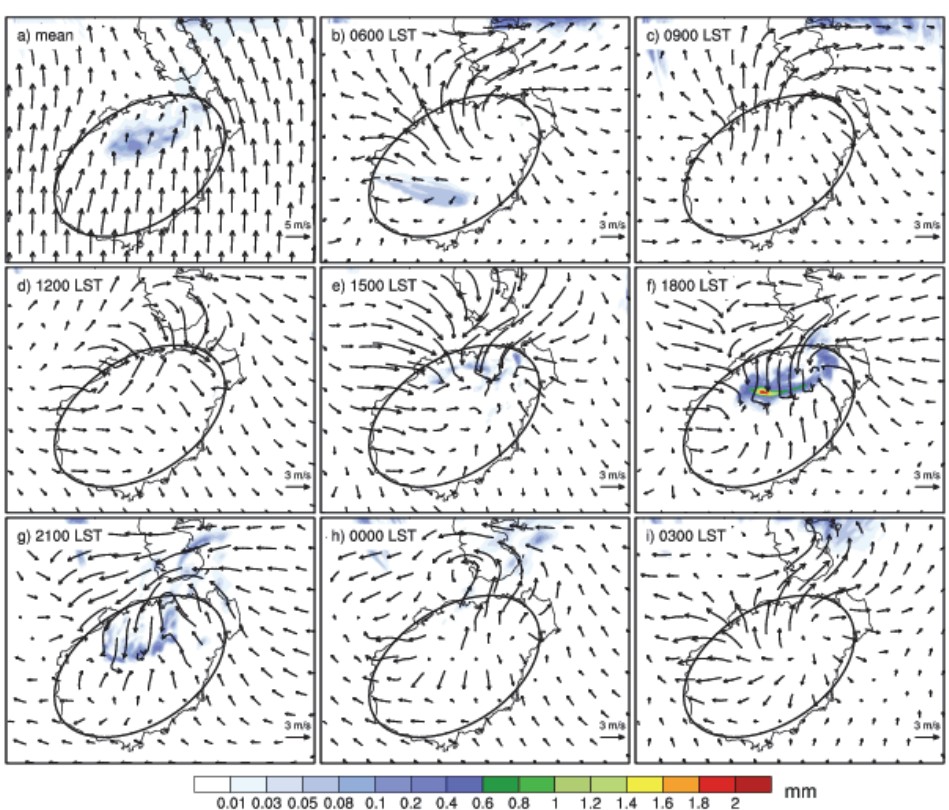

Fig. 17. As in Fig. 8, but for simulation Fakedry.



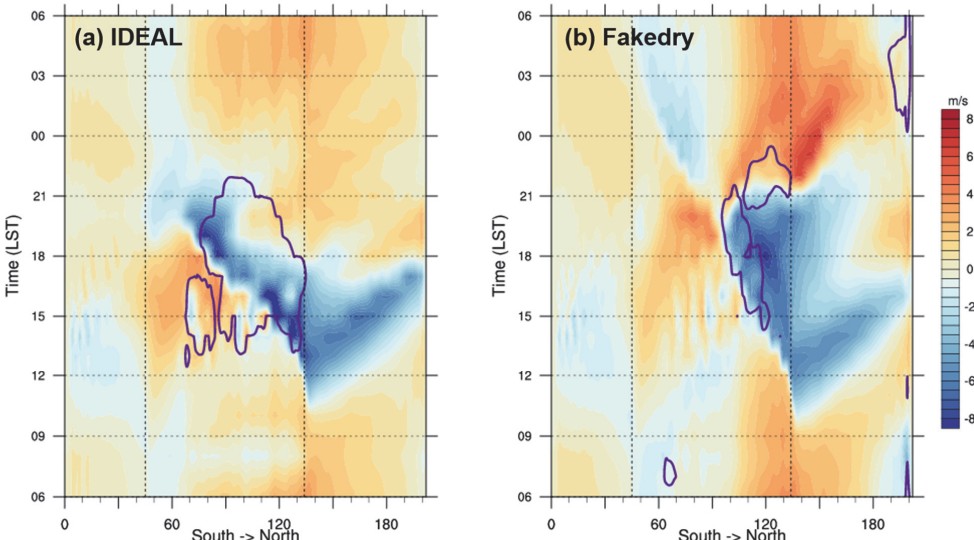

Fig. 18. Hovmoller diagrams of perturbation meridional wind component on the second lowest
model level for horizontal wind (shading) in the (a) IDEAL and (b) Fakedry simulations,
respectively. Precipitation exceeding 0.1 mm is enclosed by the heavy purple contours. The two
vertical dash lines indicate the edges of the island.





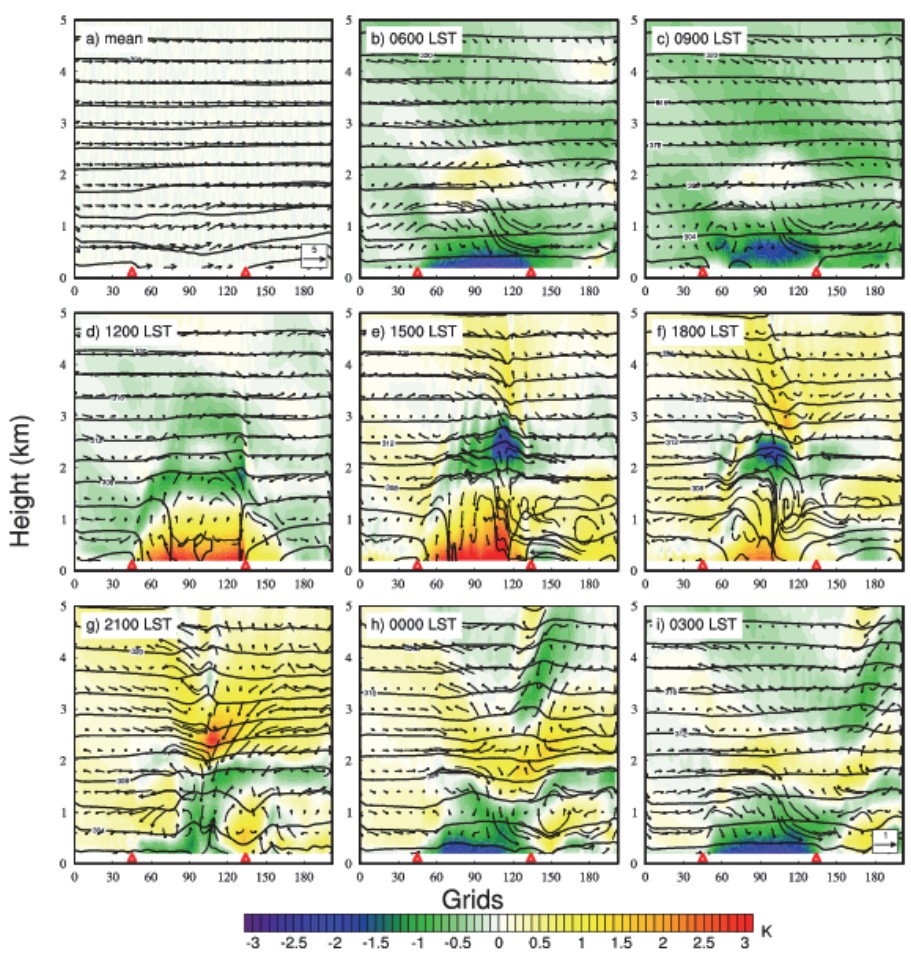

Fig. 19. As in Fig. 15, but for simulation Fakedry.