# Peer review of "The influence of sea- and land-breeze circulations on the diurnal variability of precipitation over a tropical island 3 Lei Zhu1,2,3, Zhiyong Meng1\*, Fuqing Zhang2,3, Paul M. Markowski2 4 5 6 1Laboratory for Climate and Ocean-Atmosphere Studies, Department of Atmospheric and Oceanic Sciences, School of Physics, Peking University, Beijing, China 7 2Department of Meteorology and Atmospheric Science"

_Atmospheric Chemistry and Physics, 2017_

## Referee Comment (RC1) · Anonymous Referee #1 · 30 May 2017

**Review of "The influence of sea- and land-breeze circulations on the diurnal variability of precipitation over a topical island"**

Scientific significance: 3

Scientific Quality: 3

Presentation Quality: 2

**General comments**

This work analyses the observed and modelled rainfall (especially in May and June) over Hainan Island off the coast of mainland China. It uses rain gauges and CMORPH rainfall data to compare with a number of convection permitting simulations performed using WRF with different setups, from NWP style to highly idealised. As such the work fits well into the scope of ACP. As a reviewer I felt capable of providing a review due to my experience using similar numerical weather prediction models and observational data sets, specifically focussing on the generation of convective storms. However, my specialism is not in the diurnal cycle of meteorology over tropical islands. As such, while I believe this work to be of good quality I am unable to comment on its originality compared to wider literature. The results show that: (1) WRF is capable (in this setup) of replicating the important aspects of the mean diurnal cycle compared to rainfall observations, (2) That removing the orography and coastal features made little difference to the diurnal cycle during the rainiest times of the year, (3) the dominant process that produces the diurnal cycle were shown to be the sea/land breezes caused by the relative surface heating/cooling of the island compared to the surrounding ocean and (4) that evaporative cooling as part of convective systems also plays an important role in the diurnal cycle.

I think that the structure of this paper needs some work. The abstract and methodology completely omit any detail about the range of simulations undertaken or the hypothesis that is being tested. Some of this detail is found in the results section but this means that it comes as something as a surprise when reading. I also think this work could have been presented in a more succinct way. Nineteen figures is probably too many and the structure of the work means that there is a lot of skipping between figures. For a number of figures only 1 or 2 panels are referred to in the text. Specifically I am unconvinced by the use of 2 m temperature and water vapour mixing ratio. Why not use thermodynamic quantities such as equivalent and virtual potential temperature. These give information about air masses (including temperature and moisture) and buoyancy and could still have important temperature and water vapour contours over plotted as needed. Also, from appearances it seems like NCL has been used for the creation of most figures making the calculation of such variables easy using pre-written scripts such as "wrf_eth" and "wrf_virtual_temp".

The description of the processes that cause specific features is also lacking in some cases. If the authors have not tested what is driving the production of those features it should be

made clear, if they have then say so. Generally the written English in the manuscript reads well, however, there are a few instances when the wording is slightly odd or does not conform with standard scientific usage. For example simulations are referred to as being "convection-allowing" when it is more usual to describe them as being "convection-permitting". While the intent is clear it should be changed to conform with previously published work. I also found the use of acronyms to be confusing "LSB" could just be referred to as "land-breeze", "DP" could just be "diurnal precipitation" etc.

**Specific comments**

**Abstract**

**23 and throughout** Change "convection-allowing" to "convection permitting".

**24-26** Change to "ERA-interim reanalysis. The simulations have a slight overestimation of rainfall amounts and a 1-h delay in peak rainfall time. The diurnal cycle of precipitation is driven by the occurrence of moist convection around noon owing to low-level convergence associated with the sea-breeze circulation"

**29** Change to "Generally precipitation dissipates quickly in the evening due to …"

**Introduction**

Be much clearer about the novel nature of your work. You have cited many other pieces of work that look into similar processes and use similar models, what do you do that hasn't been done elsewhere.

**49** What grid-spacing and model did Hassim et al., (2016) use? Include it here. Hassim looks at the importance of the sea breeze in the initiation of rainfall but focusses on large-scale atmospheric properties preferential to the propagation of systems offshore at night. This is not what you have said here, much more detail is required here.

**50-51** Be specific about what was beneficial about orography and gravity waves. This is not enough detail.

**52-53** In what way was island size important, how do the findings from these papers relate to the size of Hainan island?

**57** Change to "Diurnal variability is only captured in some places and months where…"

**61** There is a long list of references here after a very vague statement, please be much more specific.

**73-76** This description of Hainan Island seems incongruous. Either remove, or, if this is part of the motivation for the work make it clear what impact your work has on Hainan and how these facts relate to that impact.

**80** replace "rather" with "more"

**87-96** Highlight why this work is unique.

**Observation dataset and methodology**

Don't describe the distribution of the gauges as homogeneous. Gauges are discrete with gaps in-between and so can never be homogeneous. Maybe just say they are "relatively evenly distributed across the island"

**100** Why are they suitable for assessing the diurnal precipitation, what is the sampling frequency? Etc.

**102** Give an idea of the time period you are talking about how many stations where built after 1951 and say what years new stations became operational. If a single station was built in 1951 and the rest were built in 2009 your current description would still be true. Be more specific!

**103-110** Please use present tense e.g. "observations are augmented" not "observations were augmented".

**105** Define NOAA and CMORPH

**105** CMORPH is a retrieval not and analysis.

**106-108** delete "as shown to be valuable in past studies of diurnal precipitation over China (e.g. He and Zhange 2010, … Zhang et al., 2014)"

**108** Change to "The CMORPH grid 0.7277° x 0.7277° (approximately 8 km by 8 km) with a temporal resolution of 30 minutes.

**111-121** You need to add a lot more detail about the specific experiments here. What version of WRF is used? Advanced Research WRF would be my assumption but it is nt specified.  A much greater amount of detail is needed about the schemes that were used with an explanation of why. If they were used to conform to Chen et al (2016) then describe the experiments performed in Chen et al (2016) too.

Each of your experiments has to be explained too, "A series of convection-allowing numerical simulations were performed" gives absolutely no detail about what you have done or why!

**Observational analysis**

**136-137**  Say that this is the monsoon season.

**149-152** Replace with "No heavy rainfall or distinct diurnal variability is observed at stations along the southern coastline (blue dots on figure 1 and blue lines on figure 2).

**149-165** What is the criteria for the different dot colours? I would assume that you would group them based on some geographical property e.g. southern coast, highlands, plain, northern coast, but it looks instead like they are grouped based on their diurnal cycle, which makes the grouping less useful.

**157** swap "also are" to "are also"

**162** change to These results indicate good agreement between CMORPH data and gauges, in particular…"

**174** change to "for the whole year and over the whole island"

**174-181** You need to highlight that the precipitation levels for panels a, b, c, d, k and l are so low that essentially the diurnal precipitation percentage is meaningless, or, even get rid of those panels.

**177-178** delete "However, the diurnal precipitation… precipitation intensity." You have just defined this, therefore this is obvious and not needed.

**178-181** Even discussing March here seems pointless, March has almost no rainfall, so the magnitude of the diurnal component of almost nothing is not interesting at all. What seems more interesting is that rainfall amounts fall from May to July, increase in August again and then drop into October. However the diurnal percentage drops steadily throughout the season. You mention "physical processes", what are these, be specific! Is it associated with synoptic scale storms, typhoons, prevailing winds? Say what and describe the processes too.

**193** do you mean the Katabatic downhill jet, (which is a component of MPSs) which would happen on both sides of mountains. MPSs (to my understanding) are a feature of the lee side of mountain or plateaus, and given the direction of the prevailing wind it is unlikely that this would have an impact on the southern edge of Hainan.

**196-198** Figure 6 discussed here is not every 3 hours as stated as panel (f) shows 1700, if this is due to this being the peak rainfall time, say this explicitly.

**Numerical simulations.**

**228-234** This is very clearly lacking a clear description of the simulations that have been conducted as part of this study and the hypotheses that are being tested by performing them. It seems like much of the detail is included later, but this structure is confusing and should be changed.

**240-241** change to "slightly higher peak values of simulated 2-meter temperature and simulated precipitation…"

**242** Say "the chosen setup of WRF-ARW has the ability to…"

**256** Evening rainfall along southern coast is missing, you should mention this, including why this might be.

**265-269** This is an experiment setup description and should not be in the results section.

**310** Fog is not the only potential reason, cold pool air could be colder, what impact does stability have on the formation of fog. It needs more detail of discussion.

**311** Im not sure that 14b does show what is stated. Maybe the colour scale of 14b is not appropriate?

**316-318** 15b and 15c don't show what you say they do, at 0900 the vectors does show land breezes at both coasts but at 0600 LST only one is clear in the figure.

**319-325** The region that at 0900 LST had suppressed 2m temperatures has elevated 2 m temperatures at 1200 LST. You do not give an explanation for such rapid warming in one part of the domain while other areas (also over land have more gradual warming). I cannot work out what the cause might be from this work.

**338** I don't believe that the deep prevailing wind is what gives that moisture pattern seen in 11e. This looks a lot more like what one would expect from low level convergence of moist air producing updraughts.

**354** Quote numbers, specifically the times at which this occurs and rates.

**361-364** Drying at 850 hPa and in cross section seems predominantly driven by downward limb of the circulation advecting dry air from aloft. Movement of moisture over the northern coast also seems to be much more likely associated with the dominant wind direction than land breezes.

**377-382** Description of simulation should not be in results.

**402-403** This statement needs investigation, how does the presence of a cold pool enhance inland penetration of the sea-breeze? What are the dynamics of the situation and what have you shown to support your assertion.

**Summary**

Summary needs to include the results above that I have said are lacking. Greater specificity and inclusion of the implications of this work in the broader context of previous and ongoing work.

**425** Boundary conditions cannot be cyclic, they come from averaged ERA-Interim data! Do you mean lateral boundary conditions.

**Figures**

Latitude and Longitude should be marked on all maps and along the cross sections.

**F2** Units should be mm hr$^{-1}$

**F3** Units cannot just be mm. This has to be a rate mm hr$^{-1}$? The caption is also not clear enough "Diurnal cycles of hourly average rainfall accumulations obtained from …"

**F4** This is a confusing figure given that we know that rainfall totals are inconsequential in panels a,b,c,d,k and l. More useful just to show months May-October when some rain actually falls. Also Caption and text refers to percentage, I think the values are not expressed as such given the range between 0.1 and 0.95.

**F5** Units should be mm hr$^{-1}$

**F6** Units should be mm hr$^{-1}$ and numbers on scale are vertically squashed.

**F7** Get rid of the horizontal mean lines, they are not very useful and make the plots more confusing.

**F8,9,10,11,12,13,14,15,16,17 and 19** The comparison between (f) panels with F6 is at a different time. This doesn't have to change but it should be made clearer in the text that this discrepancy in comparison is present.

**F13** seems almost entirely pointless as a figure. It seems like the authors have tried out some new visualisation software and were very keen to include a figure using the resultant images without considering what it is that they were trying to show with such an image. It is difficult to interpret and the colours on the only panel referred to (b) are almost impossible to distinguish.

**F18** Need to include information about approximate height above the surface of the second model level, both in caption and in main text.

---

## Referee Comment (RC2) · Anonymous Referee #2 · 5 Jun 2017

The present study investigates the diurnal variation of precipitation over Hainan, an island in South China Sea. The authors document the rainfall climatology over Hainan using long-time observational records from gauge and satellite (CMORPH), and manage to show convincingly that dynamical and physical processes on the diurnal time scale are the primarily contributor to rain climatology. Mechanisms of diurnal rainfall are examined using a set of well-designed cloud-permitting numerical simulations driven by the ERA-interim reanalysis data. Results from these numerical experiments indicate that precipitation diurnal cycle is mostly due to land-sea breeze, whereas the island orography is of secondary importance. These diagnosis and modeling results are new and deserved to be published. The manuscript is overall well structured and

presented. I recommend minor revision for publication in ACP.

Major comments:

1. A numerical sensitivity experiment (FakeDry) is used to demonstrate the impact of cold pool on the sea breeze. In this FakeDry run, all latent heating and cooling is turned off. This prevents both diabatic feedback from the latent heat of condensation in the whole troposphere and cold pool due to rain reevaporation in the lowest 1-2 km. Both can be responsible for the difference between FackDry and the control run. So, conclusions (e.g., line 28-29, Line 446-449) from this FakeDry experiment regarding the role of cold pool may be revised. Otherwise, another experiment turning off rain re-evaporation in the lowest 1-2 km may be conducted to further clarify the exclusive roles of cold pool versus diabatic heating throughout the troposphere.

Specific comments:

Line 45: it is stated that "precipitation is usually due to convection". What else could rain come from other than convection?

Line 101: "full" records?

Lines 112-113: What are the surface boundary conditions? Is surface temperature predicted over both land and sea, or just predicted over land? What is the scheme for land processes?

Lines 178-179: it is stated that "The precipitation is extremely light in March and somewhat heavy in September". This statement needs some corroborating evidence, as none of the figures shows diagnostics of precipitation intensity.

Line 208: as -> at?

Lines 245-246, 328-330: Here surface temperature decrease is attributed to precipitation and cold pool. From the surface energy budget point of view, surface temperature is controlled by a range of processes: surface heat fluxes, both shortwave and long-

wave radiative processes, diffusion in the soil, etc. It is at least equally likely that decrease of surface temperature may be attributed to decreases of incoming solar heating and persistent longwave cooling.

Lines 393-394: Here the discussion of cold pool may be revised since the role of diabatic heating in the whole troposphere may also be important.

Lines 398-399: It is stated that the land sea breeze circulations "are confined to lower levels owing to weaker vertical motion". Any evidence to support this statement of causality?

Lines 425: Some discussion may be needed to justify using cyclic boundary conditions since none of the flow or surface boundary condition (SST) are cyclic in the horizontal.

Lines 450-455: Model resolution may be a convenient culprit responsible for the 1-hour delay of the rainfall (which in my opinion should be not a concern). On the other hand, there can be many other factors causing this delay, for example, biases in ECMWF reanalysis data used for boundary conditions to drive the numerical simulations, biases in physical processes (microphysics, surface processes, radiative process, etc.). It is difficult to rule out these possibilities.

Figures 14,15,18, and 19: It makes more sense to label the horizontal axis with kilometers instead of grid points.

[Figure]

---

## Author Comment (AC1) · 21 Aug 2017

**Response to Comments from Referee #1**

**We thank this reviewer for his/her detailed and insightful comments which are very helpful in our revision of the manuscript. We have made every effort to address all the concerns raised by this review and we hope our efforts will bring our manuscript closer to being accepted for publication on ACP. Our point-by-point response is given below.**

*General comments: I think that the structure of this paper needs some work.* The abstract and methodology completely omit any detail about the range of simulations undertaken or the hypothesis that is being tested. Some of this detail is found in the results section but this means that it comes as something as a surprise when reading. I also think this work could have been presented in a more succinct way. Nineteen figures are probably too many and the structure of the work means that there is a lot of skipping between figures. For a number of figures only 1 or 2 panels are referred to in the text. Specifically, I am unconvinced by the use of 2 m temperature and water vapour mixing ratio. Why not use thermodynamic quantities such as equivalent and virtual potential temperature. These give information about air masses (including temperature and moisture) and buoyancy and could still have important temperature and water vapour contours over plotted as needed. Also, from appearances it seems like NCL has been used for the creation of most figures making the calculation of such variables easy using pre-written scripts such as "wrf_eth" and "wrf_virtual_temp".

R: We thank the reviewer for the detailed suggestions above, which are mostly adopted in the revision. The description of all experiments has now been included in the methodology part, and the abstract has been revised to highlight the key findings of this work. We removed Figs. 15, 16, 17, 19 for the succinctness of the main manuscript. We chose to use 2-m temperature and water vapor mixing ratio instead of equivalent and virtual potential temperature because these potential variables may compound the impacts from radiative heating/cooling versus sources of moisture and transport which we seek to distinguish.

The description of the processes that cause specific features is also lacking in some cases. If the authors have not tested what is driving the production of those features it should be made clear, if they have then say so. Generally, the written English in the manuscript reads well, however, there are a few instances when the wording is slightly odd or does not conform with standard scientific usage. For example, simulations are referred to as being "convection-allowing" when it is more usual to describe them as being "convection permitting". While the intent is clear it should be changed to conform with previously published work. I also found the use of acronyms to be confusing "LSB" could just be referred to as "land-breeze", "DP" could just be "diurnal precipitation" etc.

R: We added more descriptions of specific physical processes in the revised manuscript per recommendation of the reviewer. All "convection-allowing" have now been revised as "convection permitting" throughout the manuscript. We also reduced the use of acronyms for a better readability, for example, "DP" was revised as "diurnal precipitation". We chose to keep the "LSB" as an acronym of "land sea breeze" because it was mentioned too many times and has been used in many papers. For example,

Lo, J.C., Lau, A.K., Fung, J.C. and Chen, F., 2006. Investigation of enhanced cross-city transport and trapping of air pollutants by coastal and urban land-sea breeze circulations. *Journal of Geophysical Research: Atmospheres*, *111*(D14).

Chen, T.C., Yen, M.C., Tsay, J.D., Liao, C.C. and Takle, E.S., 2014. Impact of afternoon thunderstorms on the land–sea breeze in the Taipei Basin during summer: an experiment. Journal of Applied Meteorology and Climatology, 53(7), pp.1714-1738.

**Specific comments**

**Abstract**
**23 and throughout** Change "convection-allowing" to "convection permitting".

**R: Changed as suggested.**

**24-26** Change to "ERA-interim reanalysis. The simulations have a slight overestimation of rainfall amounts and a 1-h delay in peak rainfall time. The diurnal cycle of precipitation is driven by the occurrence of moist convection around noon owing to low-level convergence associated with the sea-breeze circulation"

**R: Revised as suggested in lines 27-30.**

**29** Change to "Generally precipitation dissipates quickly in the evening due to …"

**R: Revised as suggested in lines 32-34.**

**Introduction**
Be much clearer about the novel nature of your work. You have cited many other pieces of work that look into similar processes and use similar models, what do you do that hasn't been done elsewhere.

**R: The novelty of this current work is that semi-idealized convection-permitting simulations with climatological mean initial conditions and diurnally averaged periodic lateral boundary conditions were used for the first time to study the dynamic and thermodynamic processes (and the impacts of land-sea breeze circulations) that control the rainfall distribution and climatology over a tropical island. This uniqueness has been clarified in the revised introduction.**

**49** What grid-spacing and model did Hassim et al., (2016) use? Include it here. Hassim looks at the importance of the sea breeze in the initiation of rainfall but focusses on large-scale atmospheric properties preferential to the propagation of systems offshore at night. This is not what you have said here, much more detail is required here.

**R: The model and grid spacing used by Hassim et al. (2016) have been added here.  We also revised the way we cited Hassim et al (2016) in a clearer way as "Hassim et al. (2016) examined the diurnal cycle of rainfall over New Guinea with a 4-km convection-allowing WRF model. They looked at the importance of the sea breeze in the initiation of rainfall but focused on large-scale atmospheric properties preferential to the propagation of systems offshore at night. They also found that**

**orography and the coastline along with gravity waves were beneficial for the longevity and maintenance of the convection systems though they were not the fundamental reason for the convection initiation." In Lines 56-62.**

**50-51** Be specific about what was beneficial about orography and gravity waves. This is not enough detail.

**R: More detailed description was added here. Please also refer to our response to the last comment.**

**52-53** In what way was island size important, how do the findings from these papers relate to the size of Hainan island?

**R: This sentence has been removed as our work does not examine impacts of different island sizes.**

**57** Change to "Diurnal variability is only captured in some places and months where…"

**R: Revised as suggested in lines 68-69.**

**61** There is a long list of references here after a very vague statement, please be much more specific.

**R: We have revised this sentence for clarity and removed some of references that were not quite relevant as "Studies show that the LSB may have different contributions to the diurnal variabilities of precipitation at different places (Keenan et al. 1988; Qian 2008; Wapler and Lane 2012; Chen et al. 2017). Precipitation tends to be initiated by the convergence of land breezes (Wapler and Lane 2012) and sea breezes (Qian 2008) over gulf area and islands area, respectively. Interactions between land breeze and prevailing wind are likely to produce precipitation over the coast area or tropical islands (Keenan et al. 1988; Chen et al. 2017)." in lines 70-75.**

**73-76** This description of Hainan Island seems incongruous. Either remove, or, if this is part of the motivation for the work make it clear what impact your work has on Hainan and how these facts relate to that impact.

**R: Those sentences have been removed as suggested.**

**80** replace "rather" with "more"

**R: Revised as suggested in line 91.**

**87-96** Highlight why this work is unique.

**R: We have highlighted the novelty as " This is the first time using semi-idealized convection-permitting simulations with climatological mean initial conditions and diurnally averaged periodic lateral boundary conditions were used for the first time to study the dynamic and thermodynamic**

**processes (and the impacts of land-sea breeze circulations) that control the rainfall distribution and climatology over a tropical island.". We would also highlight the removal of terrain effect, along with the further simplification using a perfect oval-shaped island. In Lines 99-104.**

**Observation dataset and methodology**
Don't describe the distribution of the gauges as homogeneous. Gauges are discrete with gaps in-between and so can never be homogeneous. Maybe just say they are "relatively evenly distributed across the island"

**R: As suggested, this sentence has been changed to "The gauges are relatively evenly distributed across the island." in lines 115-116.**

**100** Why are they suitable for assessing the diurnal precipitation, what is the sampling frequency? Etc.

**R:  The sampling frequency is one hour, which is dense enough to represent the diurnal rainfall cycle over the island. We have clarified this in the revised manuscript in lines 116-117.**

**102** Give an idea of the time period you are talking about how many stations where built after 1951 and say what years new stations became operational. If a single station was built in 1951 and the rest were built in 2009 your current description would still be true. Be more specific!

**R:  A table is added in the revised manuscript to show the observation period.**

**Table 1  Information of stations used in this work**

| Station No. | Latitude | Longitude | Height (m) | Obs period(YearMonth) | Name |
|---|---|---|---|---|---|
| 59757 | 20 | 110.37 | 9.9 | 197701-201212 | Qiongshan |
| 59758 | 20 | 110.25 | 63.5 | 195101-201212 | Haikou |
| 59838 | 19.1 | 108.62 | 7.6 | 195506-201212 | Dongfang |
| 59842 | 19.9 | 109.68 | 31 | 196201-201212 | Lingao |
| 59843 | 19.73 | 110 | 31.4 | 195901-201212 | Dengmai |
| 59845 | 19.52 | 109.58 | 169 | 195505-201212 | Zanzhou |
| 59847 | 19.27 | 109.05 | 98.1 | 196605-201212 | Changjiang |
| 59848 | 19.23 | 109.43 | 215.6 | 196201-201212 | Baisha |
| 59849 | 19.03 | 109.83 | 250.9 | 195602-201212 | Qiongzhong |
| 59851 | 19.7 | 110.33 | 24.2 | 196301-201212 | Dingan |
| 59854 | 19.37 | 110.1 | 118.3 | 196301-201212 | Tunchang |
| 59855 | 19.23 | 110.47 | 24 | 195509-201212 | Qiaonghai |
| 59856 | 19.62 | 110.75 | 21.7 | 195901-201212 | Wenchang |
| 59940 | 18.75 | 109.17 | 155 | 196202-201212 | Ledong |
| 59941 | 18.77 | 109.52 | 328.5 | 196301-201212 | Wuzhishan |
| 59945 | 18.65 | 109.7 | 68.6 | 196509-201212 | Baoting |
| 59948 | 18.22 | 109.58 | 419.4 | 196201-201212 | Sanya |
| 59951 | 18.8 | 110.33 | 39.9 | 196201-201212 | Wanning |
| 59954 | 18.5 | 110.03 | 13.9 | 195601-201212 | Lingshui |

**103-110** Please use present tense e.g. "observations are augmented" not "observations were augmented".

**R: The tense has been revised to present tense in line 122.**

**105** Define NOAA and CMORPH

**R: NOAA and CMORPH have been defined as "National Oceanic and Atmospheric Administration (NOAA) and Climate Prediction Center Morphing Technique (CMORPH)" in lines 122-123.**

**105** CMORPH is a retrieval not an analysis.

**R: "...the CMORPH analyses" was changed to "...the CMORPH data" as suggested in line 123.**

**106-108** delete "as shown to be valuable in past studies of diurnal precipitation over China (e.g. He and Zhang 2010, … Zhang et al., 2014)"

**R: Removed as suggested.**

**108** Change to "The CMORPH grid with a temporal resolution of 30 minutes.

**R: Revised as suggested in line 125.**

**111-121** You need to add a lot more detail about the specific experiments here. What version of WRF is used? Advanced Research WRF would be my assumption but it is not specified. A much greater amount of detail is needed about the schemes that were used with an explanation of why. If they were used to conform to Chen et al (2016) then describe the experiments performed in Chen et al (2016) too. Each of your experiments has to be explained too, "A series of convection-allowing numerical simulations were performed" gives absolutely no detail about what you have done or why!

**R: The Advanced Research WRF (ARW) 3.7.1 was used in this study, which has been cited in lines 127-128. Detailed description of all numerical simulations was included in the revised manuscript in lines 133-138 and Lines 142-154. We admit that the choice of WRF schemes is subjective in nature, based primarily on many years of experience of the senior authors, and we did not perform exclusively sensitivity tests to the choice of numerous combinations of different WRF physical parameterization schemes, nor did Chen et al. (2016). However, the realistic rainfall distribution simulated by the control simulation in comparison to climatology gives us confidence that the simulations so designed are well suited for this study.**

**Observational analysis**
**136-137** Say that this is the monsoon season.

**R: As suggested, the sentence "Most of the precipitation falls from April to October and exhibits a distinct diurnal cycle during that period" has changed to "Most of the precipitation falls from April to**

**October, which is the monsoon season, and exhibits a distinct diurnal cycle during that period" in lines 166-168.**

**149-152** Replace with "No heavy rainfall or distinct diurnal variability is observed at stations along the southern coastline (blue dots on figure 1 and blue lines on figure 2).

**R: Replaced as suggested in lines 182-184.**

**149-165** What is the criteria for the different dot colours? I would assume that you would group them based on some geographical property e.g. southern coast, highlands, plain, northern coast, but it looks instead like they are grouped based on their diurnal cycle, which makes the grouping less useful.

**R: We grouped the stations based on their similarity in diurnal variations, which to a large extent are correlated with their geographical locations so they are internally consistent. We chose to keep the station coloring unchanged. We have included this reason in lines 180-182.**

**157** swap "also are" to "are also"

**R: Revised as "were also" in line 189.**

**162** change to "These results indicate good agreement between CMORPH data and gauges, in particular ..."

**R: Revised as suggested in lines 194-195.**

**174** change to "for the whole year and over the whole island"

**R: Revised as suggested in line 205.**

**174-181** You need to highlight that the precipitation levels for panels a, b, c, d, k and l are so low that essentially the diurnal precipitation percentage is meaningless, or, even get rid of those panels.

**R: Even though low precipitation occurs during those months, their diurnal cycles can still be important because the small diurnal precipitation amount can be potentially important for plant and vegetation, and air quality. We prefer to keep them for completeness of the annual evolution of diurnal cycles. We nevertheless focus on the May and June months that have the highest diurnal precipitation amount for the rest of the study.**

**177-178** delete "However, the diurnal precipitation... precipitation intensity." You have just defined this, therefore this is obvious and not needed.

**R: Deleted as suggested.**

**178-181** Even discussing March here seems pointless, March has almost no rainfall, so the magnitude of the diurnal component of almost nothing is not interesting at all. What seems more interesting is that rainfall amounts fall from May to July, increase in August again and then drop into October. However the diurnal percentage drops steadily throughout the season. You mention "physical processes", what are these, be specific! Is it associated with synoptic scale storms, typhoons, prevailing winds? Say what and describe the processes too.

**R: For the reasons discussed above, we choose to keep the discussion despite the small amount of total precipitation in March. We do keep the discussion on the low precipitation month brief and focus primarily on May and June months.**

**Besides the processes listed by the reviewer (synoptic scale storms, typhoons, prevailing winds), other possible mechanisms include changes in the land sea temperature contrast, atmospheric moisture content, which will be explored in our future study. We added a brief discussion giving these physical processes in Lines 213-217.**

**193** do you mean the Katabatic downhill jet, (which is a component of MPSs) which would happen on both sides of mountains. MPSs (to my understanding) are a feature of the lee side of mountain or plateaus, and given the direction of the prevailing wind it is unlikely that this would have an impact on the southern edge of Hainan.

**R: The mountain plain solenoid circulation has been revised as downhill jet circulation in line 229.**

**196-198** Figure 6 discussed here is not every 3 hours as stated as panel (f) shows 1700, if this is due to this being the peak rainfall time, say this explicitly.

**R: The sentence "(every 3 h in Fig. 6)" has been revised as "(every 3 h except using 1700 LST as it is the peak rainfall time in CMORPH observation in Fig. 6)" in lines 232-233.**

**Numerical simulations.**
**228-234** This is very clearly lacking a clear description of the simulations that have been conducted as part of this study and the hypotheses that are being tested by performing them. It seems like much of the detail is included later, but this structure is confusing and should be changed.

**R: All the description of the simulations has been moved to the observation dataset and methodology section.**

**240-241** change to "slightly higher peak values of simulated 2-meter temperature and simulated precipitation…"

**R: Changed as suggested in lines 269-270.**

**242** Say "the chosen setup of WRF-ARW has the ability to…"

**R: Revised as suggested in lines 271-272.**

**256** Evening rainfall along southern coast is missing, you should mention this, including why this might be.

**R: As suggested, we added a sentence in Lines 286-289 as "However, the evening rainfall along southern coast is missing, which is likely because the resolution we used in this study is still too coarse to resolve the convection over the southern coast or the physical process was so complex that the model is unable to represent it for now".**

**265-269** This is an experiment setup description and should not be in the results section.

**R: The description of the experimental setup has been moved to the observation dataset and methodology section.**

**310** Fog is not the only potential reason, cold pool air could be colder, what impact does stability have on the formation of fog. It needs more detail of discussion.

**R: We agree with the reviewer that fog may not be the only potential reason. Other alternative possible reasons are also given here as "The slower warming in the northeastern part of the Hainan Island is likely due to the morning fog or cold pool air (Fig. 13b) that commonly forms within the area humidified (Fig. 14b) by late-afternoon precipitation on the preceding day. The cloud over the area attenuates solar radiation and subsequently slows the local warming. Moreover, positive horizontal temperature advection exists over the southern island (Fig. 15), which helps to increase the temperature faster over the area." (Lines 340-345). The impact of stability on the formation of fog is not quite relevant to the discussion here and thus not included in the revision.**

[Figure]

**Fig. 15. Horizontal temperature advection (shaded) and horizontal wind (vector) on the first model level.**

**311** I'm not sure that 14b does show what is stated. Maybe the colour scale of 14b is not appropriate?

**R:  This figure has been revised with relative humidity (RH) replacing water vapor mixing ratio. Much moister air can be easily observed during the evening and early morning, which is beneficial for the generation of morning fog, cold pool air or others.  Below shows the revised Figure 14.**

[Figure]

Fig. 14. Vertical cross-sections of relative humidity (shading), perturbation wind (vectors;the scale of the vertical component is increased by a factor of 5), and temperature (contours) in the south-to-north direction (see red line in Fig. 1) averaged over all hours (a) and at 3-h intervals (b–i). The triangles in each panel indicate the edges of the island.

**316-318** 15b and 15c don't show what you say they do, at 0900 the vectors does show land breezes at both coasts but at 0600 LST only one is clear in the figure.
**R: We agree that our original description is not clear. The sentence has been revised as "Two land-breeze circulations (LBCs) appear clearly in the vertical direction below 3 km along the coast of the island at 0600 LST (Fig. 14b). The southern LBC recedes quickly with the reversal of the temperature gradient at around 0900 LST, while the other LBC remains distinct (Fig. 14c)." in lines 348-351. We added arrows marking the locations of the LBCs (below shows the new Fig. 14 with the LBCs in b and c).**

[Figure]

Fig. 14. Vertical cross-sections of relative humidity (shading), perturbation wind (vectors; the scale of the vertical component is increased by a factor of 5), and temperature (contours) in the south-to-north

direction (see red line in Fig. 1) averaged over all hours (a) and at 3-h intervals (b–i). The triangles in each panel indicate the edges of the island.

**319-325** The region that at 0900 LST had suppressed 2m temperatures has elevated 2 m temperatures at 1200 LST. You do not give an explanation for such rapid warming in one part of the domain while other areas (also over land have more gradual warming). I cannot work out what the cause might be from this work.

R: The unevenly warming temperature over the island is a result of the horizontal temperature advection, which was demonstrated in new Figure 15 showing horizontal temperature advection with the horizontal wind on the lowest model level. From the evening to the early morning (2100-0600 LST), the temperature over the island is lower than the surrounding ocean, then the temperature along the south coastlines will increase faster than the other area as affected by positive horizontal temperature advection. After 0900 LST, the temperature over the island is warmer than the surrounding ocean, so the temperature will increase slower along the coast as affected by the negative horizontal temperature advection. It is just the opposite phenomenon along the north coastline areas that affected by negative temperature advection during the evening to early morning and positive temperature advection at other times. This reason and the new Figure 15 have been added in the revised manuscript in Lines 352-364.

[Figure]

**Figure 15. Horizontal temperature advection (shaded) and horizontal wind (vector) on the first model level.**

**338** I don't believe that the deep prevailing wind is what gives that moisture pattern seen in 11e. This looks a lot more like what one would expect from low level convergence of moist air producing updraughts.

**R: We agree. The sentence has been revised as "Moisture air is transported from ocean to the island persistently by the deep southwesterly prevailing wind throughout the lower troposphere (Fig.11e) while low level convergence of moist air generates strong updraughts, which results in an increase of the moisture over the midlevels (Fig. 14e)." in lines 377-380.**

**354** Quote numbers, specifically the times at which this occurs and rates.

**R: We revised the sentence as "The rate of temperature decrease is fastest in the first several hours during this stage, reaching around -1.5 K/h during 1800-1900 LST (red line in Fig. 7d)" in lines 396-397.**

**361-364** Drying at 850 hPa and in cross section seems predominantly driven by downward limb of the circulation advecting dry air from aloft. Movement of moisture over the northern coast also seems to be much more likely associated with the dominant wind direction than land breezes.

**R: We agree. The drying at 850 hPa was attributed to the downward flow as stated in lines 406-408. Movement of moisture over the northern coast was also attributed to the dominant wind direction than land breezes as stated in Lines 408-410.**

**377-382** Description of simulation should not be in results.

**R: Description of simulation has been moved to "Observation dataset and methodology" section.**

**402-403** This statement needs investigation, how does the presence of a cold pool enhance inland penetration of the sea-breeze? What are the dynamics of the situation and what have you shown to support your assertion.

**R: When a cold pool moves toward the island/inland, the pressure gradient should increase which drives the flow more inland.**

**Summary**
Summary needs to include the results above that I have said are lacking. Greater specificity and inclusion of the implications of this work in the broader context of previous and ongoing work.

**R: The conclusion part has been revised as suggested. In particular, main findings as summarized by this reviewer in his/her general comment are summarized. (1) WRF is capable (in this setup) of replicating the important aspects of the mean diurnal cycle compared to rainfall observations, (2) That removing the orography and coastal features made little difference to the diurnal cycle during the rainiest times of the year, (3) the dominant process that produces the diurnal cycle were shown to be the sea/land breezes caused by the relative surface heating/cooling of the island compared to the surrounding ocean and (4) that evaporative cooling as part of convective systems also plays an important role in the diurnal cycle. Those have been concluded in lines 488-494.**

**425** Boundary conditions cannot be cyclic, they come from averaged ERA-Interim data! Do you mean lateral boundary conditions?

**R: Thanks for this correction. The "cyclic boundary conditions" has been revised as "cyclic lateral boundary conditions" in lines 481-482.**

**Figures**

Latitude and Longitude should be marked on all maps and along the cross sections.

**R: All the map and cross section figures have been added with latitude and longitude.**

**F2** Units should be mm hr-1

**R: The units has been revised as "mm h$^{-1}$"**

**F3** Units cannot just be mm. This has to be a rate mm hr-1? The caption is also not clear enough "Diurnal cycles of hourly average rainfall accumulations obtained from …"

**R: The units has been revised as "mm h$^{-1}$", and the caption has been revised as "Diurnal cycles of hourly average rainfall accumulations obtained from gauges (blue) and CMORPH (red) in each month".**

**F4** This is a confusing figure given that we know that rainfall totals are inconsequential in panels a,b,c,d,k and l. More useful just to show months May-October when some rain actually falls. Also Caption and text refers to percentage, I think the values are not expressed as such given the range between 0.1 and 0.95.

**R:  As we discussed in response above, even though the rainfall is quite small, the percentage of the diurnal precipitation is still quite important. We choose to keep those panels. The caption has been revised as "Percentage of the total precipitation that can be attributed to the diurnal cycle…..".**

**F5** Units should be mm hr-1

**R: Revised as suggested.**

**F6** Units should be mm hr-1 and numbers on scale are vertically squashed.

**R: The units has been revised as "mm h$^{-1}$". The numbers on scale were revised to have a normal look.**

**F7** Get rid of the horizontal mean lines, they are not very useful and make the plots more confusing.

**R: Removed as suggested.**

**F8,9,10,11,12,13,14,15,16,17 and 19** The comparison between (f) panels with F6 is at a different time. This doesn't have to change but it should be made clearer in the text that this discrepancy in comparison is present.

**R: We have clarified this discrepancy in the text.**

**F13** seems almost entirely pointless as a figure. It seems like the authors have tried out some new visualisation software and were very keen to include a figure using the resultant images without considering what it is that they were trying to show with such an image. It is difficult to interpret and the colours on the only panel referred to (b) are almost impossible to distinguish.

**R:  We chose to keep this figure to give a good big picture of the relationship between multiple variables including the 2-m temperature, wind perturbation, cloud and precipitation feature.**

**F18** Need to include information about approximate height above the surface of the second model level, both in caption and in main text.

**R: The height if the second model level has been added in the caption and main text. The second lowest model level is the 0.994-sigma level, which is about 50 m above the surface.**

**Response to Comments from Referee #2**

**We thank this reviewer for his/her detailed and insightful comments which are very helpful in our revision of the manuscript. We have made every effort to address all the concerns raised by this review and we hope our efforts will bring our manuscript closer to being accepted for publication on ACP. Our point-by-point response is given below.**

*Major comments:*
1. A numerical sensitivity experiment (FakeDry) is used to demonstrate the impact of cold pool on the sea breeze. In this FakeDry run, all latent heating and cooling is turned off. This prevents both diabatic feedback from the latent heat of condensation in the whole troposphere and cold pool due to rain re-evaporation in the lowest 1-2 km. Both can be responsible for the difference between FakeDry and the control run. So, conclusions (e.g., line 28-29, Line 446-449) from this FakeDry experiment regarding the role of cold pool may be revised. Otherwise, another experiment turning off rain re-evaporation in the lowest 1-2 km may be conducted to further clarify the exclusive roles of cold pool versus diabatic heating throughout the troposphere.

**R: Another sensitivity experiment NOVAP which turns off rain re-evaporation has been conducted as suggested.  Comparing rainfall and the propagation of sea breeze among the experiments IDEAL, FakeDry and NOVAP, we found that the propagation speed in NOVAP is in the middle of the IDEAL and FakeDry, which means that the cold pool can speed up the propagation. Also, it can be found that the precipitation intensity is much greater in NOVAP, indicating that the cold pool can suppress the precipitation significantly. The description of NOVAP experiment has been included in the "Observation dataset and methodology" part. The results and comparison of the experiments have been included in "The impacts of latent heating/cooling on LSB and the related rainfall" part.**

**Specific comments:**
Line 45: it is stated that "precipitation is usually due to convection". What else could rain come from other than convection?

**R: The sentence has been revised as "Tropical precipitation is usually due to convection" in line 49.**

Line 101: "full" records?

**R: Revised as suggested. The sentence "though records exist ..." has been revised as "though full records exist ..." in line 118.**

Lines 112-113: What are the surface boundary conditions? Is surface temperature predicted over both land and sea, or just predicted over land? What is the scheme for land processes?

**R: The surface layer option is the revised MM5 Monin-Obukhov scheme, and the land-surface option is thermal diffusion scheme. Both land and sea surface temperatures are predicted, which has been included in lines 136-137.**

Lines 178-179: it is stated that "The precipitation is extremely light in March and somewhat heavy in September". This statement needs some corroborating evidence, as none of the figures shows diagnostics of precipitation intensity.

**R: The hourly precipitation averaged over the whole island in each month has been calculated in Fig. 3. It shows clearly that the hourly precipitation is close to zero in both CMORPH and gauges data in March, while it becomes larger than 0.5 mm/h in September.**

Line 208: as -> at?

**R: "As" was revised as "representing" in line 245.**

Lines 245-246, 328-330: Here surface temperature decrease is attributed to precipitation and cold pool. From the surface energy budget point of view, surface temperature is controlled by a range of processes: surface heat fluxes, both shortwave and long wave radiative processes, diffusion in the soil, etc. It is at least equally likely that decrease of surface temperature may be attributed to decreases of incoming solar heating and persistent longwave cooling.

**R: Many other reasons that may have contributed to the temperature decrease have been included here. This sentence has been revised as "decreases rapidly thereafter owning primarily to the development of precipitation (which has its diurnal maximum during this period) and associated evaporative cooling. Besides, surface temperature is also controlled by other processes, such as surface heat fluxes, both shortwave and long wave radiative processes, diffusion in the soil, etc." In Lines 367-370.**

Lines 393-394: Here the discussion of cold pool may be revised since the role of diabatic heating in the whole troposphere may also be important.

**R: The discussion of cold pool has been revised based on the new experiment NOVAP in lines 434-443 as "A weaker sea breeze is observed in the FakeDry experiment than in the IDEAL experiment while NOVAP experiment shows the strongest sea breeze. The NOVAP experiment generated much stronger precipitation over the island as the rain evaporation cooling process was turned off. The propagation of the LSB is much slower and the inland propagation distance is much shorter in FakeDry experiment than that in the IDEAL experiment while the propagation speed in NOVAP experiment is between the other two experiments, which suggests that cold pool can accelerate the over the tropical island. In the NOVAP experiment, the heavy precipitation does not dissipate after 2100 LST but propagates into the northeast out of the island with the land breeze, indicating that cold pool plays an important role in dissipating the convection."**

Lines 398-399: It is stated that the land sea breeze circulations "are confined to lower levels owing to weaker vertical motion". Any evidence to support this statement of causality?

**R: As the latent heating and cooling processes were turned off, the advection in FakeDry was much weakened. At the same time, the sea breeze decreased. In term of land breeze at their peak stage, the strong land breeze circulation can reach around 2 km in vertical altitude in IDEAL experiment**

while it is lower than 1 km in FakeDry experiment (Fig. R1h, i vs. Fig. R2h, i). For sea breeze at their peak stage, the sea breeze circulation can extend to 5 km altitude in IDEAL experiment, while it only can reach 3 km in FakeDry experiment (Fig. R1e,f vs. Fig. R2e,f). The stronger vertical velocity (negative and positive) can extend over 5 km in IDEAL experiment, while it can only reach 2 km in FakeDry experiment. We added this evidence in the revised manuscript in Lines 448-449. Figs. R1 and R2 are the original Figs. 15 and 19. For the sake of succinctness, we removed these two figures in the revised manuscript.

[Figure]

Fig. R1 Vertical cross-sections of perturbation temperature (shading), perturbation wind (vectors; the scale of the vertical component is increased by a factor of 5), and temperature (contours) in the south-to-north direction (see red line in Fig. 1) averaged over all hours (a) and at 3-h intervals (b–i) in simulation IDEAL. The triangles in each panel indicate the edges of the island.

[Figure]

Fig. R2 Vertical cross-sections of perturbation temperature (shading), perturbation wind (vectors; the scale of the vertical component is increased by a factor of 5), and temperature (contours) in the south-to-north direction (see red line in Fig. 1) averaged over all hours (a) and at 3-h intervals (b–i) in simulation FakeDry. The triangles in each panel indicate the edges of the island.

Lines 425: Some discussion may be needed to justify using cyclic boundary conditions since none of the flow or surface boundary condition (SST) are cyclic in the horizontal.

**R: Sorry for the confusion. We actually used the cyclic lateral boundary conditions rather than cyclic boundary conditions.**

Lines 450-455: Model resolution may be a convenient culprit responsible for the 1-hour delay of the rainfall (which in my opinion should be not a concern). On the other hand, there can be many other factors causing this delay, for example, biases in ECMWF reanalysis data used for boundary conditions to drive the numerical simulations, biases in physical processes (microphysics, surface processes, radiative process, etc.). It is difficult to rule out these possibilities.

**R: More possible reasons for the 1-hour delay have been included in the summary part in lines 518-521.**

Figures 14,15,18, and 19: It makes more sense to label the horizontal axis with kilometers instead of grid points.

**R: We have changed all the horizontal axis label to latitude and longitude.**

---

## Author Response (AR2)

**Response to Comments from Referee #1**

**We thank this reviewer for his/her detailed and insightful comments which are very helpful in our revision of the manuscript. We have made every effort to address all the concerns raised by this review and we hope our efforts will bring our manuscript closer to being accepted for publication on ACP. Our point-by-point response is given below.**

This work describes the seasonal and diurnal cycle of rainfall over Hainan island in the South China sea. Firstly it uses both rain gauge station data and satellite retrievals (CMORPH) of rainfall to describe the seasonal changes in rainfall associated with the arrival and retreat of the Asian monsoon as well as analysing the proportion of rainfall occurring due the the diurnal cycle across the year. The peak rainfall period is then analysed further using Weather Research and Forecasting (WRF) simulations of varying idealisations. The simulations highlight the roles of different meteorological features and ultimately determine the importance of the sea- and land-breeze circulations and the production of downdraughts from convective storms in the production of the diurnal cycle of rainfall.

The manuscript has been modified significantly since I first reviewed it. There are fewer figures, the methodology has become clearer and results are discussed in a more coherent manner. This has greatly improved the way in which this work reads and emphasises the findings of the work much more than in its previous incarnation. As such I believe that this work is publishable after only a few minor technical corrections.

**Specific recommendations**

14-20 Sentence too long. Maybe change line 17 to "…as well as numerical simulations. The simulations are the first to use climatological …"
**R: Revised as suggested.**

18 delete "periodic", the boundaries are not periodic as features propagating out of one side are not forced to appear in the lateral boundary conditions on the other side of the domain.
**R: As suggested, we revised the statement as "The simulations are the first to use climatological mean initial and lateral boundary conditions to study the …". Similar changes were made in Lines 101 and 480.**

23 delete "as well"
**R: Deleted as suggested.**

27 change to "simulations have a slight overestimation…"
**R: Revised as suggested.**

49 change to "(Dai 2001), tropical convection is also well…"

**R: Revised as suggested.**

73-74 change to "gulf and island areas respectively."
**R: Revised as suggested.**

81 change to "This work is…"
**R: Revised as suggested.**

103 change to "We also highlight…"
**R: Revised as suggested.**

141 do you need to put 0000 UTC in twice?
**R: The last 0000 UTC may be not necessary. This sentence has been revised as "cycled from 0000 to 0600, 1200, and 1800 UTC."**

197-202 What you define is not a percentage. It needs to be multiplied by 100 to be a percentage, this is clear in the range in figure 4 being 0 to 1, not 0 to 100.
**R: We changed all "percentage" to "fraction" throughout the manuscript. Thank you very much for this correction.**

277 -278 I am not convinced that they look very similar, certainly diurnal behaviour is similar but the mean values look quite different!
**R: We meant that the pattern of the precipitation distribution was generally similar. For example, high precipitation were mainly located in the northeast part of the island (Fig. R1). We agree that the mean values are somewhat different. This sentence was revised as "The horizontal distribution of precipitation averaged in REAL (Fig. 8a), especially the location of the high precipitation area, also has reasonably good agreement with that of the CMORPH data at all hours (Fig. 6a), although the simulated precipitation is somewhat larger."**

[Figure]

**Fig. R1    The mean precipitation of CMORPH (a) and REAL experiment (b),**

**adapted from Fig. 6a and Fig. 8a. The two red contours denote the high precipitation area in CMORPH.**

320-321 add numbers before each stage e.g. "The stages are: (1) the establishment of a sea breeze (0600-1200 LST), (2) the peak sea breeze…"
**R: As suggested, the relevant sentences were revised as "The stages are: (1) the establishment of a sea breeze (0600−1200 LST), (2) the peak sea breeze and peak precipitation (1200−1800 LST), (3) the establishment of a land breeze (1800−0000 LST), and (4) the peak land breeze phase (0000−0600 LST)."**

449 It is more usual to say "not shown" rather than "figures omitted".
**R: Revised as "not shown".**

458 change to "over tropical areas is poorly…"
**R: Revised as suggested.**

465 change to "Most precipitation falls during the warm season…"
**R: Revised as suggested.**

467 change to "Precipitation is at a maximum…"
**R: Revised as suggested.**

472 change to "The analysis of CMORPH data shows that…"
**R: Revised as suggested.**

485-486 change to "Even with an idealized elliptical and flat island, located at…"
**R: This sentence was revised as "Even with an idealized elliptical and flat island covered by only grassland, located at the same place with similar area and orientation, the diurnal cycle characteristics can still be fairly well captured."**

**Response to Comments from Referee #2**

**We thank this reviewer for his/her detailed and insightful comments which are very helpful in our revision of the manuscript. We have made every effort to address all the concerns raised by this review and we hope our efforts will bring our manuscript closer to being accepted for publication on ACP. Our point-by-point response is given below.**

I have one minor technical suggestion for the revised manuscript. It is stated in Lines 481-482 that the lateral boundary conditions are cyclic and they are generated using 10-year (2006–2015) average of ERA-interim data. However, one can either use cyclic conditions or climatology at lateral boundaries, but not both. It seems that using climatology is more appropriate here since the numerical domains are located at subtropical latitudes.

**R: We removed "cyclic" in Line 480. Similar changes were made in Lines 18 and 101.**